EMBO
Molecular Medicine

# Macrophages reprogramming improves immunotherapy of IL-33 in peritoneal metastasis of gastric cancer

Keying Che [1,2,3], Yuting Luo[1], Xueru Song[1], Zhe Yang[3], Hanbing Wang[1], Tao Shi [1], Yue Wang[1], Xuan Wang[1,4], Hongyan Wu[5], Lixia Yu[1], Baorui Liu[1] & Jia Wei [1,2,6,7] ✉

## Abstract

**Peritoneal metastasis (PM) has a suppressive tumor immune microenvironment (TIME) that limits the effects of immunotherapy. This study aimed to investigate the immunomodulatory effects of intraperitoneal administration of IL-33, a cytokine that is reported to potentiate antitumor immunity and inhibit metastasis. We found survival was significantly prolonged in patients with high IL-33 mRNA expression. In immunocompetent mice, intraperitoneal administration of IL-33 could induce a celiac inflammatory environment, activate immunologic effector cells, and reverse the immunosuppressive tumor microenvironment, which effectively delayed tumor progression and PM of gastric cancer. Mechanistically, IL-33 could induce M2 polarization by activating p38-GATA-binding protein 3 signaling. IL-33 combined with anti-CSF1R or p38 inhibitor to regulate tumor-associated macrophages (TAMs) had a synergistic antitumor effect. Inducing a local inflammatory milieu by IL-33 administration provided a novel approach for treating peritoneal metastasis, which, when combined with TAM reprogramming to reshape TIME, can achieve better treatment efficacy.**

**Keywords** IL-33; Gastric Cancer; Peritoneal Metastasis; Tumor Immune Microenvironment; Tumor-associated Macrophages
**Subject Categories** Cancer; Digestive System; Immunology

## Introduction

Gastric cancer (GC) is the sixth most common cancer and the third leading cause of cancer-related deaths globally, resulting in 1 million newly diagnosed cases and an estimated 800,000 deaths yearly (Sung et al, 2021). Peritoneal metastasis (PM) is the most common form of recurrence and metastasis in patients with GC

(Dong et al, 2019). Approximately, 53–66% of patients diagnosed with distant metastatic GC present with PM, which is associated with a dismal 5-year survival rate of 6.0% (Rau et al, 2020). Although surgery and chemotherapy can improve the clinical prognosis of patients, the commonly used therapeutic measures for GC with PM are still unsatisfactory. Recent advances in genomic sequencing and molecular profiling have revealed several promising therapeutic targets and elucidated novel biology, particularly the role of the surrounding tumor immune microenvironment (TIME) in PM of GC (Gwee et al, 2022).

PM is limited to the abdominal cavity; therefore, the local administration of therapeutic agents can achieve higher drug concentrations with a few systemic adverse effects. In recent years, intraperitoneal chemotherapy and local cytokine therapy have been explored in peritoneal tumors (Bonnot et al, 2019; Iwamura et al, 2017). Interleukin (IL)-33 is a cytokine of the IL-1 superfamily that triggers a MyD88-dependent acute inflammatory response through its surface receptor ST2 (Kita, 2022). MyD88 binding recruits IL-1 receptor-associated kinase and tumor necrosis factor receptor-associated factor 6-activating nuclear factor–kappa B (NF-κB) pathway, which promote inflammatory cytokine expression. Meanwhile, IL-33 signaling has been demonstrated to enhance the expression of forkhead box-P3 and GATA-binding protein 3 (GATA3) by enhancing transforming growth factor β1 (TGF-β1)-mediated differentiation through a p38-dependent mechanism (Griesenauer and Paczesny, 2017). Previous studies have indicated that IL-33 predominantly induces a type 2 immune response and is closely associated with allergic and parasitic diseases (Chan et al, 2019). It has recently been reported that IL-33 potentiates antitumor immunity by promoting inflammatory cell infiltration and function. Recently, an increasing number of published studies have shown that IL-33 can recruit and activate CD8[+] T cells and natural killer (NK) cells to inhibit the metastasis of tumor cells (Luo et al, 2020; Qi et al, 2020). Considering the spatial limitations of PM, we investigated the local abdominal administration of IL-33 to activate the TIME and enhance antitumor immunity.

Macrophages are important innate immune cells involved in an inflammatory response and tumor progression (Stengel et al, 2020).

[1]Department of Oncology, Nanjing Drum Tower Hospital, Affiliated Hospital of Medical School, Nanjing University, Nanjing, China. [2]State Key Laboratory of Pharmaceutical Biotechnology, School of Life Sciences, Nanjing University, Nanjing, China. [3]Tumor Research and Therapy Center, Shandong Provincial Hospital Affiliated to Shandong First Medical University, Jinan, China. [4]Department of Pathology, The Third Affiliated Hospital of Soochow University, Changzhou, China. [5]Department of Pathology, Nanjing Drum Tower Hospital, Affiliated Hospital of Medical School, Nanjing University, Nanjing, China. [6]Chemistry and Biomedicine Innovation Center (ChemBIC), Nanjing University, Nanjing, China. [7]Engineering Research Center of Protein and Peptide Medicine, Ministry of Education, Nanjing, China. ✉E-mail: jiawei99@nju.edu.cn

Increasing evidence suggests that IL-33 acts as a complex and dual-role cytokine, which is crucial in initiating the immune response and differentiating macrophages (Faas et al, 2021; Qi et al, 2020). However, the molecular mechanisms underlying IL-33-driven polarization of macrophages remain poorly understood. Revealing the molecular mechanisms of IL-33-induced polarization of macrophages may contribute to better therapeutic effects based on IL-33.

This study demonstrated that the intraperitoneal administration of IL-33 could induce the celiac inflammatory environment, activate immunologic effector cells, and reverse the immunosuppressive TIME, which delayed tumor progression and PM of GC.

## Results

### Differential expression of IL-33 and its effect on prognosis in patients with GC

We downloaded the RNA sequencing (RNA-seq) data from TCGA-STAD, GTEx, and GEO data sets to evaluate the expression of *IL-33* mRNA in normal tissues, tumor tissues, and tumor cells derived from malignant ascites. The expression of *IL-33* mRNA decreased in purified tumors from ascites compared with normal and primary tumor tissues (Fig. 1A). In addition, the expression of *IL-33* mRNA in human GC cell lines was evaluated using qRT-PCR. The *IL-33* mRNA expression of seven human GC cell lines was extremely low compared with the positive control (normal gastric tissue), indicating that the expression of *IL-33* was low in GC (Appendix Fig. S1A). The median overall survival (OS) time was 37.93 months in patients with GC having high expression of *IL-33* mRNA and 25.20 months in patients having low expression of *IL-33* mRNA ($P = 0.011$). The median progression-free survival time was 25.90 months in the *Il-33* mRNA high-expression group and 14.10 months in the *Il-33* mRNA low-expression group ($P = 0.014$) (Fig. 1B).

Furthermore, 170 patients with biopsy-proven GC were enrolled from the Cancer Center of Nanjing Drum Tower Hospital between August 2017 and January 2019. The clinicopathological characteristics of the cohorts are summarized in Appendix Table S1 (Dataset EV1). H-scores were used to examine IL-33 levels in formalin-fixed paraffin-embedded (FFPE) tissue samples of these patients using immunohistochemical (IHC) staining (Fig. 1C). Higher expression of IL-33 protein was observed in normal tissues (black arrows) than in tumor tissues (red arrows). We found 42 IHC sections with normal tissues. The staining intensity of IL-33 in normal tissues was higher than that in tumor tissues ($P = 0.0476$) (Fig. 1D). Patients were categorized into the IL-33 low-expression group (H-score ≤4; $n = 133$, 78.24%) and the IL-33 high-expression group (H-score >4; $n = 37$, 21.76%). The 3-year OS rate was 74.96% in patients with GC in the IL-33 protein high-expression group and 55.45% in the IL-33 protein low-expression group ($P = 0.0141$) (Fig. 1E).

The RNA-seq data of TCGA-STAD was analyzed using xCell to further explore the effect of IL-33 on the TIME of patients with GC. The results showed that the high expression of *IL-33* mRNA was associated with increased infiltration of various immune cells, such as CD8$^+$ T cells ($P < 0.0001$), DCs ($P < 0.0001$), and M2 macrophages ($P < 0.0001$). In addition, the high expression of

*IL-33* mRNA positively correlated with immune score ($P < 0.0001$), stroma score ($P < 0.0001$), and microenvironment score ($P < 0.0001$), suggesting that the high expression of *IL-33* mRNA was associated with the activation of the TIME (Fig. 1F).

The platelet-to-lymphocyte ratio (PLR), neutrophil-to-lymphocyte ratio (NLR), lymphocyte-to-monocyte ratio (LMR), and C-reactive protein (CRP) level were obtained to evaluate the correlation between IL-33 expression and inflammatory response. No significant correlation was found between blood inflammatory markers and the percentage of positive cells (Appendix Fig. S1B) or H-score (Appendix Fig. S1C) of IL-33, suggesting that locally high expression of IL-33 might not trigger a strong inflammatory response.

### IL-33 delayed tumor growth in mice with abdominal dissemination of GC

Considering the observed low expression of *IL-33* mRNA in the tumor cells derived from malignant ascites, we further evaluated the antitumor effect of IL-33 in abdominal dissemination tumors. To this end, we established an abdominal dissemination tumor model by challenging 615-line mice with mouse forestomach carcinoma (MFC) cells and then treating them with mIL-33 or phosphate-buffered saline (PBS) (Fig. 2A). We found a significant decrease in abdominal dissemination tumors and mesenteric dissemination tumors in the IL-33 treatment group (Fig. 2B). The tumor weight ($P = 0.0032$) and the number of mesenteric dissemination tumor nodules ($P = 0.0022$) of mice with IL-33 treatment also decreased (Fig. 2C). In addition, we observed the survival of mice during treatment. The results showed that the median survival time of mice in the vehicle group was 23 days, while that of mice in the IL-33 treatment group could be extended to 30 days; a statistically significant difference in survival time was observed between the two groups ($P = 0.0373$) (Appendix Fig. S2A). The ascites in the two groups were collected on D16. In the vehicle group, one of the five mice had no ascites and one had a small amount of ascites that could not be removed. In the IL-33 treatment group, only three of the five mice had ascites. The degree of bloody ascites was more obvious in the vehicle group compared with the IL-33 treatment group, indicating that IL-33 reduced the tumor load in the abdominal cavity of mice (Appendix Fig. S2B). No obvious toxicity or alteration in body weight was observed in the experimental group (Appendix Fig. S2C,D). The aforementioned results indicated that the local intraperitoneal administration of IL-33 could effectively trigger the antitumor effect of the abdominal dissemination of GC.

### IL-33 promoted the infiltration and activation of peritoneal immunocytes and induced local inflammatory milieu

The tumor and spleen were collected and analyzed by flow cytometry at the treatment endpoint to evaluate the influence of IL-33 on the TIME. The results showed that abdominal tumors had significantly higher infiltration of CD8$^+$ T cells ($P = 0.0004$) (Fig. 2D), NK cells ($P = 0.0016$) (Fig. 2E), DCs ($P = 0.0018$) (Fig. 2F), macrophages ($P = 0.0190$) (Fig. 2G), and M2 macrophages ($P = 0.0126$) (Fig. 2H) in the IL-33 treatment group

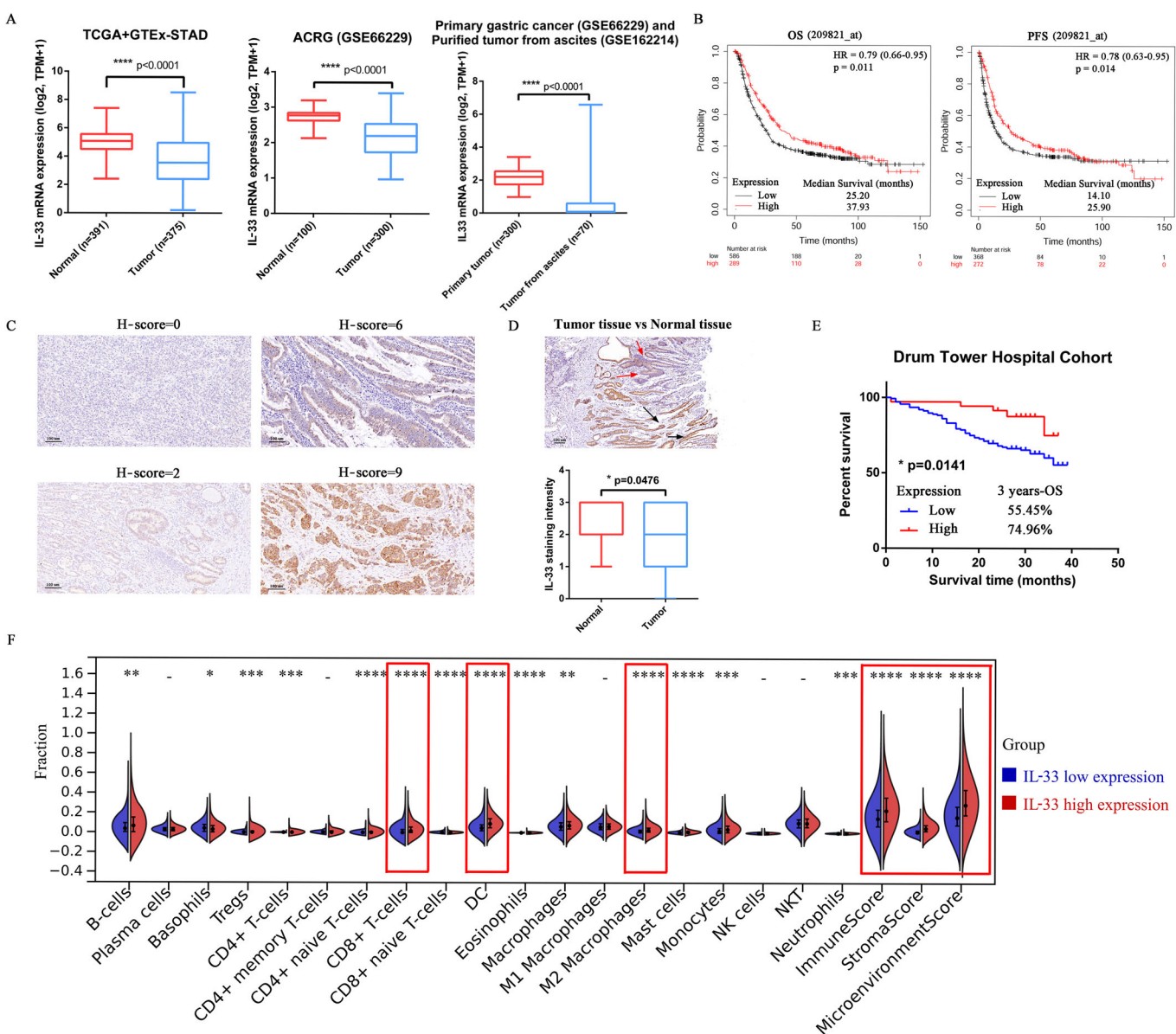

**Figure 1. Differential expression of IL-33 and its effect on prognosis in patients with GC.**

(A) IL-33 mRNA expression (expressed in transcripts per million) of normal tissues, primary tumor tissues, and purified tumors from ascites obtained from TCGA, GTEx, and GEO databases ($n = 391$ in Normal group of TCGA+GTEx-STAD, $n = 375$ in Tumor group of TCGA+GTEx-STAD, $n = 100$ in Normal group of ACRG, $n = 300$ in Tumor group of ACRG, $n = 300$ in Primary tumor group, $n = 70$ in Tumor from ascites group, biological replicates). (B) Kaplan–Meier survival curves of OS and PFS for patients with low or high expression of IL-33 in GC. (C) GC tissues ($n = 170$) from the Drum Tower Hospital Cohort were stained for IL-33. The representative images with different $H$-scores are shown (scale bars = 100 μm). (D) Representative image and quantitative analysis of IL-33 staining intensity in normal and tumor tissue areas. The black arrows show normal gastric glands, and the red arrows show cancerous glands ($n = 42$ in Normal group, $n = 170$ in Tumor group, biological replicates). (E) Kaplan–Meier survival curves of OS in patients with low ($n = 135$) or high ($n = 35$) IL-33 expression in the Drum Tower Hospital Cohort. (F) Fractions of immune cell infiltration in GC tissues from TCGA-STAD were estimated using xCell ($n = 203$ in IL-33 low expression group, $n = 311$ in IL-33 high expression group, biological replicates). Data information: The data with error bars are shown as mean ± SD. ns, not significant. *$P < 0.05$, **$P < 0.01$, ***$P < 0.001$, ****$P < 0.0001$, as determined using the two-tailed unpaired-sample Student $t$ test (A,D), logrank test (B,E), or Wilcoxon rank-sum test (F). Also, see Appendix Fig. S1. Source data are available online for this figure.

---

compared with the vehicle group. No significant difference was observed in the infiltration of M1 macrophages between the two groups (Fig. 2I). Moreover, the level of interferon (IFN)-γ of CD8$^+$ T cells ($P = 0.0004$) (Fig. 2J), NK cells ($P < 0.0001$) (Fig. 2K), and CD4$^+$ T cells ($P < 0.0001$) (Fig. 2L) also increased. The aforementioned

results indicated that the antitumor effect of immune cells was activated by IL-33.

Further, we also evaluated the changes in immunocyte subgroups in the spleen after the intraperitoneal administration of IL-33. The proportion of CD8$^+$ T cells ($P = 0.0262$) (Appendix Fig. S3A) and the

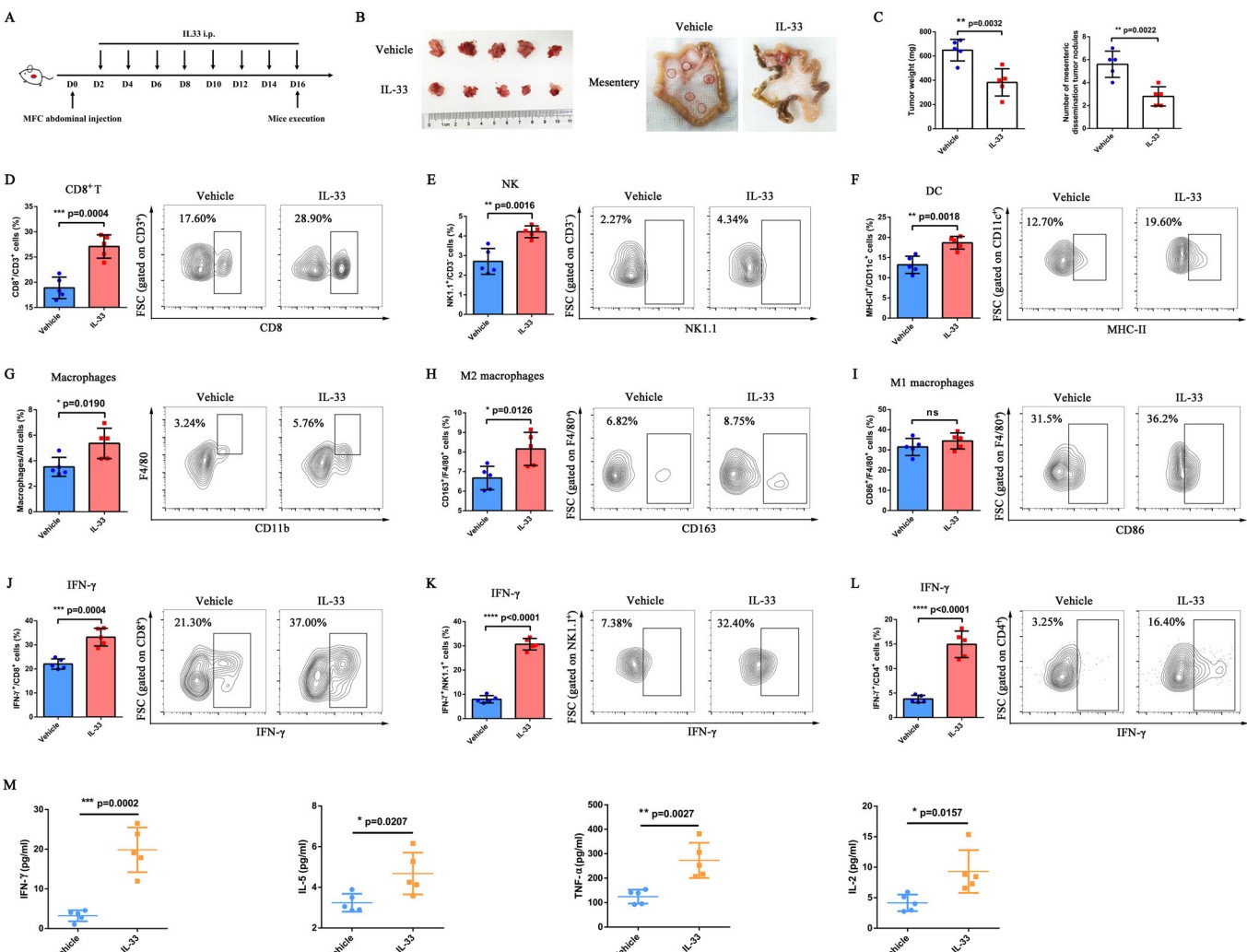

**Figure 2. IL-33 delayed tumor growth, promoted the infiltration and activation of peritoneal immunocytes, and induced local inflammatory milieu.**

(A) Schematic illustration of IL-33 treatment in the MFC abdominal dissemination model. 615-line mice ($n = 5$ per group) were injected intraperitoneally with $5 \times 10^5$ MFC cells and treated intraperitoneally with PBS or IL-33 every other day. PBS-treated mice served as controls. (B) Abdominal and mesenteric dissemination tumors of MFC-challenged 615 mice treated with PBS or IL-33. (C) Tumor weight and number of mesenteric dissemination tumor nodules in MFC-challenged 615 mice treated with PBS or IL-33 ($n = 5$ biological replicates). (D) Proportions of CD8+/CD3+ cells in abdominal tumors were determined by flow cytometry ($n = 5$ biological replicates). (E) Proportions of NK1.1+/CD3− cells in abdominal tumors were determined by flow cytometry ($n = 5$ biological replicates). (F) Proportions of MHC-II+/CD11c+ cells in abdominal tumors were determined by flow cytometry ($n = 5$ biological replicates). (G) Proportions of macrophages/all cells in abdominal tumors were determined by flow cytometry ($n = 5$ biological replicates). (H) Proportions of CD163+/F4/80+ macrophages in abdominal tumors were determined by flow cytometry ($n = 5$ biological replicates). (I) Proportions of CD86+/F4/80+ macrophages in abdominal tumors were determined by flow cytometry ($n = 5$ biological replicates). (J) Proportions of IFNγ+/CD8+ cells in abdominal tumors were determined by flow cytometry ($n = 5$ biological replicates). (K) Proportions of IFNγ+/NK1.1+ cells in abdominal tumors were determined by flow cytometry ($n = 5$ biological replicates). (L) Proportions of IFNγ+/CD4+ cells in abdominal tumors were determined by flow cytometry ($n = 5$ biological replicates). (M) Expression of IFNγ, IL-5, TNF-α, and IL-2 in ascites of the abdominal dissemination mouse model was detected using CBA ($n = 5$ biological replicates). Data information: The data with error bars are shown as mean ± SD. ns, not significant; *$P < 0.05$, **$P < 0.01$, ***$P < 0.001$, ****$P < 0.0001$, as determined by the two-tailed unpaired-sample Student $t$ test. Also, see Appendix Figs. S2, S3. Source data are available online for this figure.

IFN-γ secretion of spleen NK cells ($P = 0.0059$) (Appendix Fig. S3B) were significantly higher in the IL-33 treatment group compared with the vehicle group. However, no significant difference was observed in the proportion of NK cells (Appendix Fig. S3C), DCs (Appendix Fig. S3D), macrophages (Appendix Fig. S3E), M1 macrophages (Appendix Fig. S3F), and M2 macrophages (Appendix Fig. S3G), and IFN-γ secretion of spleen CD8+ T cells (Appendix Fig. S3H) and CD4+ T cells (Appendix Fig. S3I) between the two groups, implying that the local intraperitoneal administration of IL-33 mainly

influenced the peritoneal TIME rather than the splenic immune microenvironment.

We collected the ascites from two groups of mice and measured Th1/Th2 cytokine levels in the supernatant of ascites using a cytometric bead array (CBA) to evaluate the effect of IL-33 on the celiac immune environment. Compared with the vehicle group, the IL-33-treated group exhibited higher levels of pro-inflammatory cytokines, such as IFN-γ ($P = 0.0002$), IL-5 ($P = 0.0207$), TNF-α ($P = 0.0027$), and IL-2 ($P = 0.0157$) (Fig. 2M). However, no differences

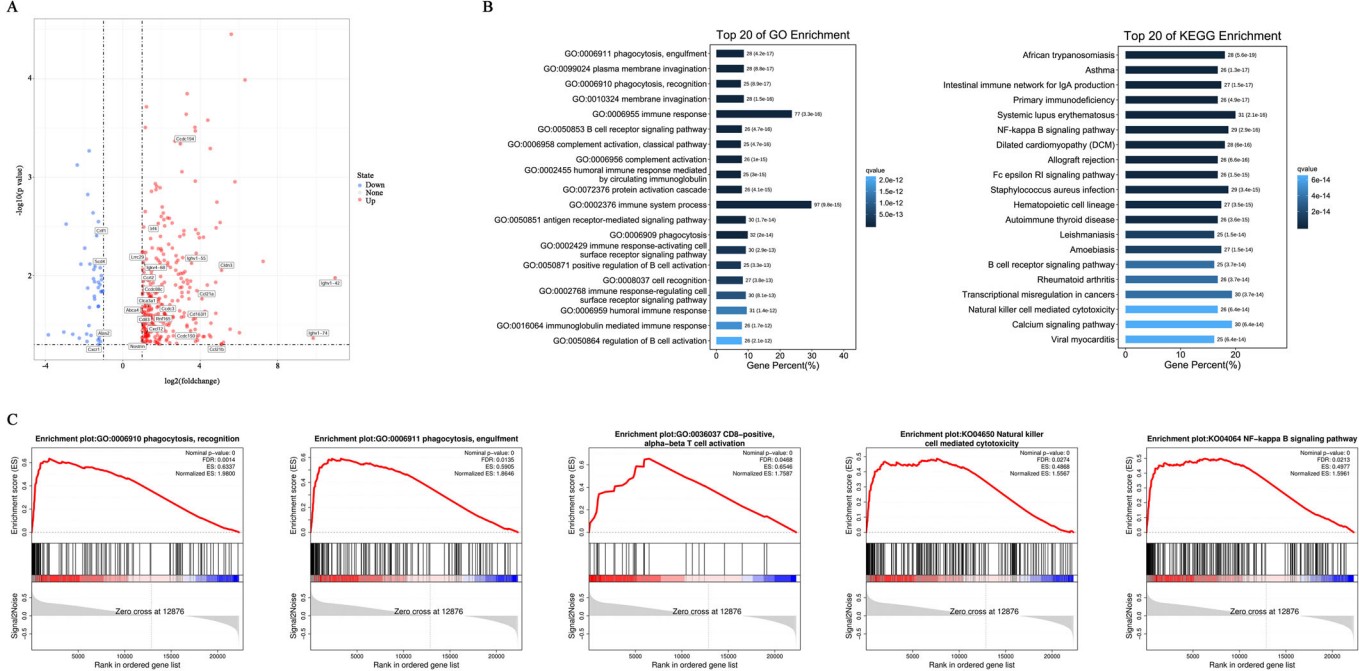

**Figure 3. IL-33 promoted macrophage recruitment in tumors and enriched phagocytosis-related pathways in the abdominal dissemination of GC.**

RNA-seq analysis was performed on the tumor samples of mice in the vehicle and IL-33-treated groups ($n = 8$ per group). (A) Volcano plot depicting transcript fold changes in tumors of mice treated with or without IL-33. Significantly downregulated and upregulated genes are painted in blue and red, respectively. (B) GO and KEGG analyses exhibited the top 20 significantly enriched pathways. The x-axis shows the gene percentage, and the y-axis shows the enriched pathway of each term. (C) Representative signaling pathways about phagocytosis and immune response in GSEA. Source data are available online for this figure.

were observed in the expression of IL-6, IL-4, IL-10, and IL-13 compared with pro-inflammatory cytokines (Appendix Fig. S3J), indicating that IL-33 could induce the local inflammatory milieu.

## IL-33 promoted macrophage recruitment in tumors and enriched phagocytosis-related pathways in the abdominal dissemination of GC

RNA-seq analysis was performed on the tumors of the vehicle and IL-33-treated mice to further evaluate the changes in related signaling pathways induced by IL-33. The volcano plot showed differentially expressed genes (DEGs), including 300 upregulated and 43 downregulated genes in the IL-33 treatment group. *Cd83* and *Cd163l1* were highly expressed in the IL-33 treatment group, suggesting increased infiltration and activation of DCs and M2 macrophages (Fig. 3A). Moreover, the phagocytosis-related and immune-response pathways were significantly enriched in mice with IL-33 administration, as detected by Gene Ontology (GO) (GO: 0006911, GO: 0006910, GO: 0006955, and GO: 0002376) and Kyoto Encyclopedia of Genes and Genomes (KEGG) (B-cell receptor signaling pathway, NK cell–mediated cytotoxicity) analyses, which was consistent with increased tumor infiltration of macrophages (Fig. 3B). In addition, we screened out the top 20 pathways based on NSE by Gene Set Enrichment Analysis (GSEA), and found that phagocytosis recognition, phagocytosis engulfment, CD8-positive alpha-beta T cell activation, NK cell–mediated cytotoxicity, and NF-κB signaling pathway were significantly enriched in the experimental group (Fig. 3C). In summary, the

data proved that the local inflammatory milieu mediated by IL-33 induced the recruitment of tumor-associated peritoneal macrophages and activation of tumor immune environment.

## IL-33 directly triggered the polarization of macrophages

The results of bioinformatic and flow cytometry analyses proved that the infiltration of M2 macrophages significantly increased after IL-33 treatment. The murine macrophage cell line RAW264.7 cells, BMDMs from 615-line mice, human monocyte cell line THP1, and human macrophages from peripheral blood mononuclear cells (PBMCs) were used for in vitro experiments to further evaluate the effects of IL-33 on the phenotype and function of macrophages. For RAW264.7 cells and BMDMs, LPS, IL-4, and IL-10 recombinant proteins were used to induce the polarization of macrophages to an M1 or M2 phenotype. For THP1 monocytes, PMA was used to induce the differentiation of macrophages. Subsequently, human macrophages were stimulated with human IFN-γ and LPS to M1 phenotype or stimulated with human IL-4 and IL-13 recombinant protein to M2 phenotype. CD80 and iNOS were detected by flow cytometry to represent M1 macrophages, whereas CD163 and CD206 were detected to represent M2 macrophages.

The positive expression rates of CD80 and iNOS significantly increased in IL-33-treated RAW264.7 cells (CD80: $P = 0.0202$, iNOS: $P = 0.0029$) (Fig. 4A). Meanwhile, the stimulation of IL-33 also upregulated the expression of CD163 and CD206 (CD163: $P < 0.0001$, CD206: $P = 0.0312$) (Fig. 4B). Moreover, the stimulation of RAW264.7 cells with IL-33increased the secretion of TNF-α

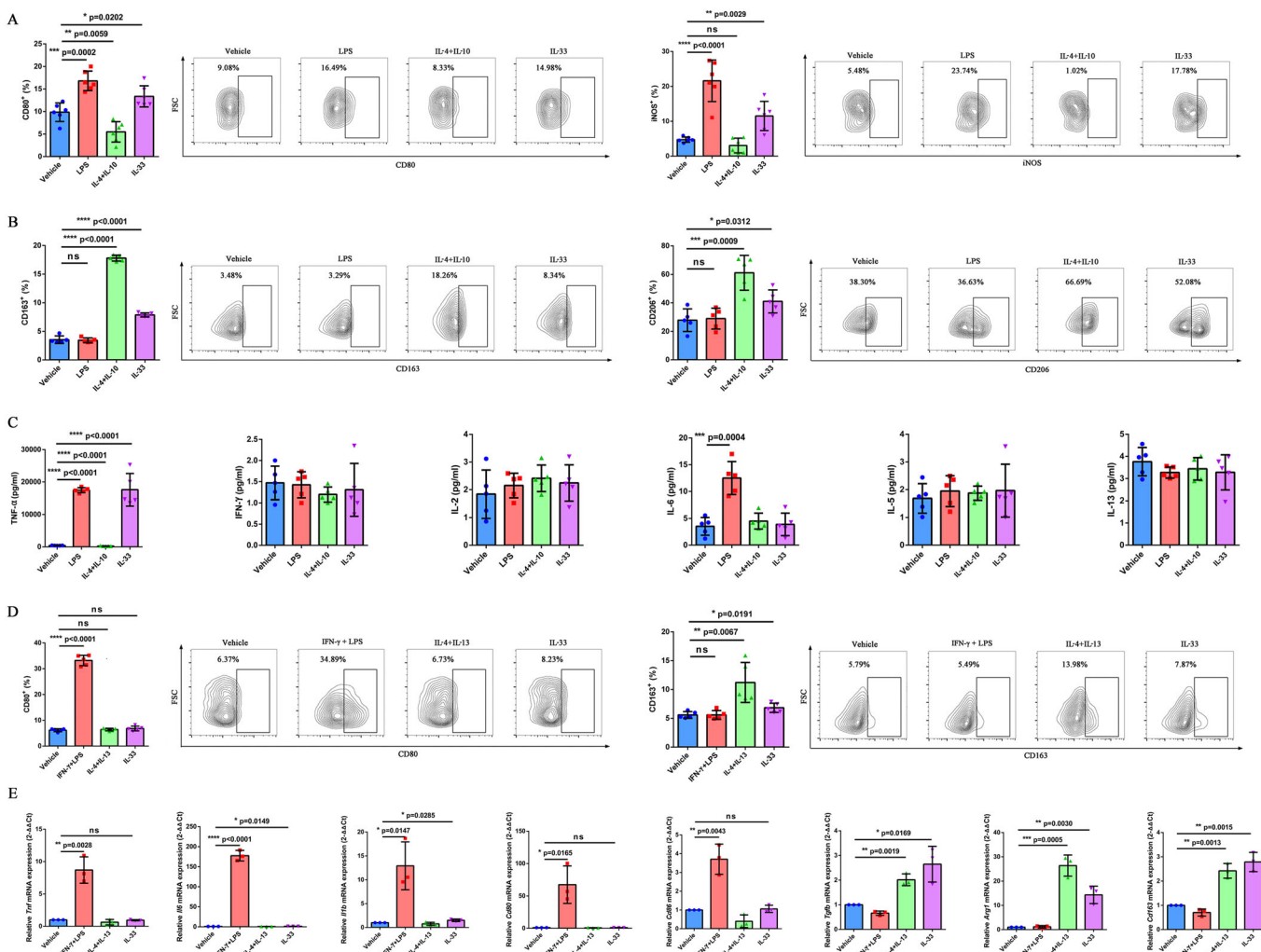

**Figure 4. IL-33 directly triggered the polarization of macrophages.**

(A) RAW264.7 cells were stimulated with PBS, LPS, IL-4 + IL-10, or IL-33 for 48 h. The proportions of CD80+ and iNOS+ RAW264.7 cells were determined by flow cytometry (n = 6 biological replicates). (B) Proportions of CD163+ and CD206+ RAW264.7 cells were determined by flow cytometry (n = 5 biological replicates). (C) Concentrations of TNF-α, IFNγ, IL-2, IL-6, IL-5, and IL-13 in the culture supernatants of RAW264.7 cells were detected using CBA (n = 5 biological replicates). (D) THP1 cells were stimulated with PBS, IFNγ + LPS, IL-4 + IL-13, or IL-33 for 48 h. The proportions of CD80+ and CD163+ THP1 cells were determined by flow cytometry (n = 5 biological replicates). (E) Relative mRNA expression of *Tnf*, *IL-6*, *IL1-β*, *Cd80*, *Cd86*, *Tgf-β*, *Arg1*, and *Cd163* in THP1 cells was evaluated using qRT-PCR (n = 3 biological replicates). Data information: The data with error bars are shown as mean ± SD. ns, not significant; *P < 0.05, **P < 0.01, ***P < 0.001, ****P < 0.0001 using two-tailed unpaired-sample Student t test. Also, see Appendix Figs. S4, S5. Source data are available online for this figure.

(P < 0.0001) (Fig. 4C). These results suggested that IL-33 could induce the phenotypic transformation of RAW264.7 into both M1 and M2 in vitro. However, the expression of M1-related markers CD80 and iNOS did not increase, while the expression of M2-related markers CD163 (P = 0.0008) and CD206 (P = 0.0001) was upregulated in IL-33-induced BMDMs compared with those in the vehicle group (Appendix Fig. S4A,B). The lack of increased expression of M1-related markers in IL-33-stimulated BMDMs might be related to the polarization of macrophages to the M2 phenotype induced by M-CSF. In addition, for THP1 macrophages, the evaluation of the expression of CD80 and CD163 by flow cytometry revealed that CD163 expression increased under IL-33 stimulation in vitro (P = 0.0191) (Fig. 4D). The stimulation of IL-33 increased the mRNA expression of *IL-6* (P = 0.0149), *IL1-β*

(P = 0.0285), *Tgf-β* (P = 0.0169), *Arg1* (P = 0.0030), and *Cd163* (P = 0.0015) (Fig. 4E). For macrophages isolated from PBMCs of patients with GC, the expression of CD80 (P = 0.0040) and CD163 (P = 0.0031) increased on IL-33 stimulation in vitro (Appendix Fig. S5A). Next, we examined the effects of IL-33 on the PBMC-mediated killing of GC cells. At low E:T ratios such as 1:1, IL-33 readily induced substantial apoptosis of SNU601 cells (P = 0.0006), suggesting that IL-33 was essential for PBMC-mediated killing of tumor cells (Appendix Fig. S5B). Together, these results confirmed that IL-33 triggered the transformation of macrophages from M0 into both M1 and M2 phenotypes. Notably, IL-33 induced the simultaneous polarization of macrophages toward M1 and M2 in vitro. However, these findings were not completely consistent with the results observed in vivo. We hypothesized that the in vivo

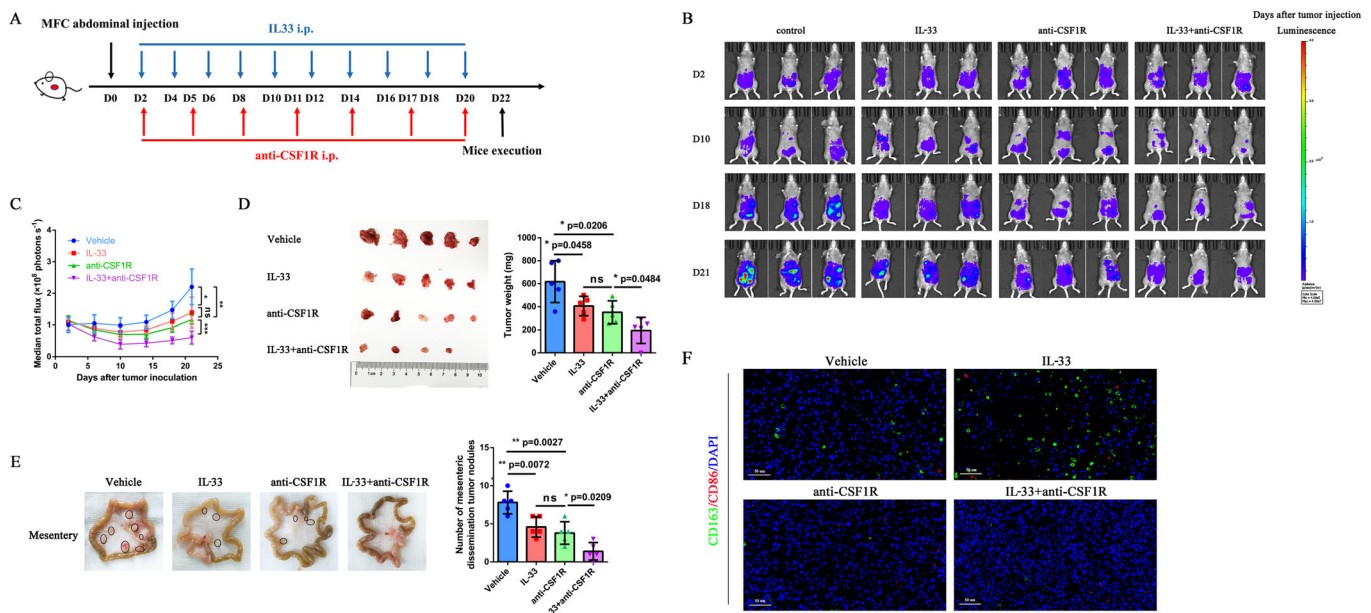

**Figure 5. Combination of IL-33 and anti-CSF1R augmented antitumor response in the MFC abdominal dissemination model.**

(A) 615-line mice ($n = 5$ per group) developing abdominal dissemination tumors upon intraperitoneal injection of MFC-Luc were treated with vehicle, IL-33, anti-CSF1R, or IL-33 + anti-CSF1R. Vehicle-treated mice served as controls. (B) Tumor growth was imaged using BLI as photons per second. (C) Tumor growth profiles of mice treated with vehicle, IL-33, anti-CSF1R, or IL-33 + anti-CSF1R ($n = 5$ biological replicates). (D) Abdominal dissemination tumors and tumor weight of MFC-Luc-challenged 615 mice treated with vehicle, IL-33, anti-CSF1R, or IL-33 + anti-CSF1R ($n = 5$ biological replicates). (E) Mesenteric dissemination tumors and the number of mesenteric dissemination tumor nodules in MFC-Luc-challenged 615 mice treated with vehicle, IL-33, anti-CSF1R, or IL-33 + anti-CSF1R ($n = 5$ biological replicates). (F) Representative images of multiplex immunofluorescence for CD86 and CD163 performed on tumor tissues of MFC-Luc-challenged 615 mice treated with vehicle, IL-33, anti-CSF1R, or IL-33 + anti-CSF1R. Nucleus (DAPI, blue); CD163 (FAM, green); CD86 (Cy5, *red*). Scale bar = 50 μm. Data information: The data with error bars are shown as mean ± SD. ns, not significant; *$P < 0.05$, **$P < 0.01$, ***$P < 0.001$ by two-tailed unpaired-sample Student $t$ test. Also, see Appendix Figs. S6, S7. Source data are available online for this figure.

TIME, especially the abdominal TIME, represented a highly immunosuppressive state. Also, multiple immunosuppressive cytokines, chemokines, and immune checkpoints were highly expressed, leading to the failure of M1 polarization induced by IL-33 in vivo.

## Combination of IL-33 and anti-CSF1R augmented antitumor response in multiple models

Macrophages account for a high proportion of immune cells in the peritoneal cavity. For an unmanipulated mouse, 30% of live cells are macrophages (Ray and Dittel, 2010). IL-33 activates immunologic effector cells and promotes the recruitment of M2 macrophages. We investigated the antitumor strategy of IL-33 combined with anti-CSF1R in multiple models to improve the antitumor efficiency. IL-33 combined with anti-CSF1R was administered simultaneously in the abdominal dissemination model of GC, and its synergistic effect against tumors was evaluated (Fig. 5A). Dynamic in vivo bioluminescence imaging (BLI) showed that IL-33 combined with anti-CSF1R effectively inhibited the abdominal dissemination of luciferase-mediated MFC cells (MFC-Luc) compared with monotherapy (Fig. 5B,C). The combined treatment group demonstrated a significant reduction in the number of abdominal and mesenteric dissemination tumors, indicating a further antitumor effect (Fig. 5D,E). The flow cytometry analysis showed that the combination therapy further increased the tumoral infiltration of CD4$^+$ T cells ($P = 0.0008$) (Appendix Fig. S6A), CD8$^+$ T cells ($P = 0.0003$) (Appendix Fig. S6B),

and NK cells ($P < 0.0001$) (Appendix Fig. S6C), and decreased the tumoral infiltration of macrophages significantly ($P < 0.0001$) (Appendix Fig. S6D) compared with those in the vehicle group, suggesting that the combination of IL-33 and anti-CSF1R could recruit antitumor immunocytes in the TIME and inhibit the filtration of TAMs. Multiplex immunofluorescence for CD86 and CD163 was performed to directly show the infiltration of macrophages in tumor tissues of 615-line mice. Immunofluorescence confirmed that IL-33 induced an accumulation of M2 macrophages, whereas CD163 was absent in tumor tissues of 615-line mice treated with anti-CSF1R (Fig. 5F). No obvious toxicity or alterations in body weight were observed in these experimental groups (Appendix Fig. S7A,B). These studies demonstrated that IL-33 triggered the accumulation of M2 macrophages in tumor tissues, and the combination of IL-33 and anti-CSF1R augmented antitumor response in the abdominal dissemination of GC.

In addition, anti-CSF1R was administered 10 days before tumor inoculation (D-10) to eliminate macrophages earlier and more thoroughly to further verify the importance of macrophages in the abdominal dissemination of GC. We observed a robust reduction in tumor growth in the anti-CSF1R and IL-33 + anti-CSF1R groups compared with the control group (Appendix Fig. S8A,B). The number of mesenteric dissemination tumors significantly reduced in the combined treatment group (Appendix Fig. S8C). The aforementioned results indicated that earlier and more thorough elimination of macrophages could achieve better antitumor effects, indicating the importance of macrophages in the progression of peritoneal dissemination.

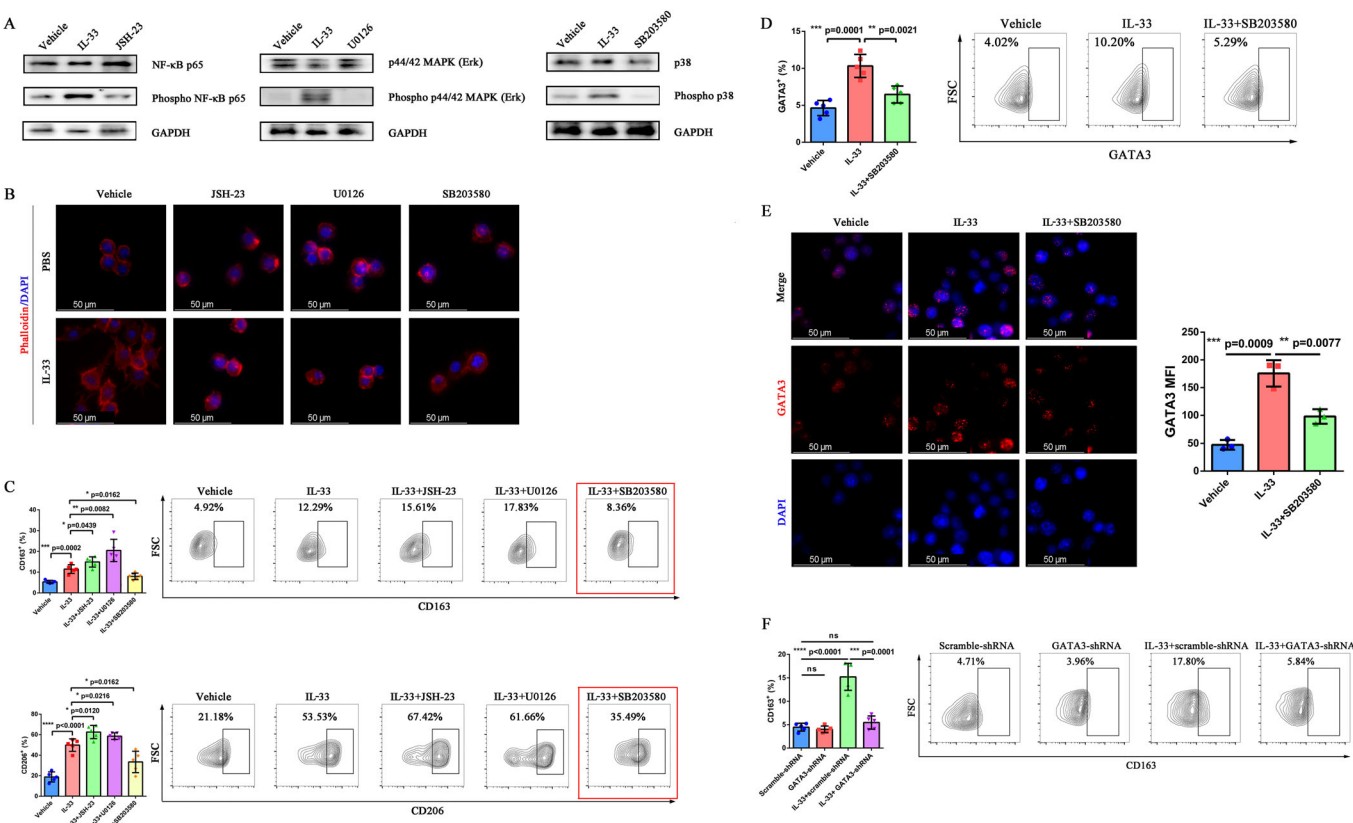

**Figure 6.  IL-33 activated the p38-GATA3 signaling pathway to induce M2 polarization of macrophages.**

RAW264.7 cells were pretreated with or without NF-κB-, p44/42-, and p38-specific inhibitors for 1 h before stimulation with IL-33 or PBS. PBS-treated cells served as controls. (**A**) Western blot analysis of NF-κB p65, phospho NF-κB p65, p44/42 MAPK, phospho p44/42 MAPK, p38, phospho p38, and GAPDH in RAW264.7 cells. (**B**) Inhibition of IL-33-induced changes in the shape of RAW264.7 cells by NF-κB-, p44/42-, and p38-specific inhibitors. Nucleus (DAPI, blue); cytoskeleton (phalloidin-TRITC, red). Scale bar = 50 μm. (**C**) Expression of CD163 and CD206 in IL-33-induced RAW264.7 cells altered by NF-κB-, p44/42-, and p38-specific inhibitors ($n = 5$ biological replicates). (**D**) Expression of GATA3 in IL-33-induced RAW264.7 cells altered by a p38-specific inhibitor ($n = 5$ biological replicates). (**E**) Representative immunofluorescence staining images and quantification of GATA3 in stimulated RAW264.7 cells. Nucleus (DAPI, blue); GATA3 (Cy5, red). Scale bar = 50 μm, $n = 3$ biological replicates. (**F**) Effects of GATA3 knockdown on CD163 expression in RAW264.7 cells ($n = 5$ biological replicates). Data information: The data with error bars are shown as mean ± SD. ns, not significant; *$P < 0.05$, **$P < 0.01$, ***$P < 0.001$, ****$P < 0.0001$ by two-tailed unpaired-sample Student $t$ test. Also, see Appendix Fig. S10. Source data are available online for this figure.

We established the subcutaneous tumor model of 615-line mice to further evaluate the antitumor effect on subcutaneous tumors. IL-33 or anti-CSF1R alone could control the tumor volume and tumor weight, whereas the combination of IL-33 and anti-CSF1R showed a further antitumor effect in the subcutaneous MFC-challenged 615-line mouse model (Appendix Fig. S9A,B). No obvious alteration in body weight was observed in these experimental groups (Appendix Fig. S9C). Hence, these data revealed that IL-33 combined with anti-CSF1R further enhanced the antitumor effect in both abdominal dissemination and subcutaneous tumor models, indicating the importance of eliminating macrophages.

## IL-33 activated the p38-GATA3 signaling pathway to induce M2 polarization of macrophages

Western blotting was performed on the critical downstream signaling pathways of IL-33, including NF-κB p65, p44/42 MAPK, and p38, to clarify the mechanism of IL-33-induced macrophage polarization. The expression levels of phospho NF-κB p65, phospho p44/42 MAPK, and phospho p38 were higher in IL-33-stimulated RAW264.7 cells compared with those in the vehicle group, suggesting that IL-33 could activate these pathways (Fig. 6A). In addition, the protein phosphorylation was inhibited by the application of related pathway inhibitors JSH-23 (NF-κB p65 inhibitor), U0126 (p44/42 MAPK inhibitor), and SB203580 (p38 inhibitor). RAW264.7 cells were stimulated with IL-33 alone or IL-33 combined with specific inhibitors to further prove the effect of IL-33 on macrophages via the aforementioned signaling pathways. The cytoskeleton was displayed by immunofluorescence to evaluate the morphological changes in cells. The results showed that IL-33 stimulation induced the morphological changes in RAW264.7 cells, which were blocked by specific inhibitors of NF-κB p65, p44/42 MAPK, and p38 (Fig. 6B).

Based on the aforementioned results, we further evaluated the expression of CD163 and CD206 on RAW264.7 cells. The expression of CD163 (IL-33 vs IL-33 + JSH-23: $P = 0.0439$; IL-33 vs IL-33 + U0126: $P = 0.0082$) and CD206 (IL-33 vs IL-33 + JSH-23: $P = 0.0120$; IL-33 vs IL-33 + U0126: $P = 0.0216$) significantly

increased when RAW264.7 cells were stimulated with IL-33 combined with U0126 or JSH-23 compared with IL-33 alone. Nevertheless, the expression of CD163 (IL-33 vs IL-33 + SB203580: $P = 0.0162$) and CD206 (IL-33 vs IL-33 + SB203580: $P = 0.0162$) decreased when the cells were stimulated with IL-33 + SB203580 compared with IL-33 alone (Fig. 6C). These results indicated that NF-κB p65 and p44/42 MAPK signaling pathways played a role in the M1 polarization of macrophages. Therefore, macrophages tended to undergo M2 polarization when these pathways were inhibited. On the contrary, activating the p38 signaling pathway promoted the M2 polarization of macrophages, and the M2 polarization tendency was blocked after inhibiting the p38 pathway.

Previous studies showed that GATA3, an IL-33-responsive transcriptional hub, was closely related to tissue repair and differentiation of macrophages (Faas et al, 2021). We evaluated the expression of GATA3 in RAW264.7 cells to elucidate the mechanism of IL-33-induced M2 polarization. The expression of GATA3 was inhibited when the cells were stimulated with IL-33 + SB203580 compared with IL-33 alone ($P = 0.0021$) (Fig. 6D). The immunofluorescence staining assay showed GATA3 level significantly increased in RAW264.7 cells stimulated with IL-33 ($P = 0.0009$). The expression of GATA3 also decreased in the IL-33 + SB203580 group compared with the IL-33-stimulated group ($P = 0.0077$) (Fig. 6E). GATA3-shRNA was constructed to knock down the GATA3 expression and activity in RAW264.7 cells to verify the role of GATA3 in M2 polarization. The expression of CD163 significantly decreased in GATA3-shRNA-transfected cells stimulated with IL-33 compared with that in the IL-33 + scramble-shRNA group ($P = 0.0001$) (Fig. 6F). These data suggested a critical role of GATA3 in macrophages that responded to an IL-33-induced reprogramming.

We further evaluated the expression of GATA3, CD80, and CD163 in human macrophages to clarify the influence of p38-GATA3 signaling pathway–induced macrophage polarization. The expression of GATA3 ($P = 0.0041$; Appendix Fig. S10A) and CD163 ($P = 0.0002$; Appendix Fig. S10B) significantly decreased when macrophages were stimulated with IL-33 + SB203580 compared with IL-33 alone. On the contrary, the expression of CD80 ($P = 0.0384$) increased when the macrophages were treated with IL-33 + SB203580 compared with IL-33 alone (Appendix Fig. S10C), suggesting that p38 inhibitor could effectively inhibit the expression of GATA3 and block the M2 polarization induced by p38-GATA3 signaling pathway.

### IL-33 combined with p38 signaling pathway inhibitor synergistically induced anti-tumor response in vivo

To evaluate the effect of p38 signaling pathway on tumor progression and macrophage polarization, we established a model of GC abdominal dissemination treated with IL-33 combined with p38 inhibitor (SB203580) and evaluated the anti-tumor efficacy of combined therapy (Fig. 7A). Compared with monotherapy group, combined therapy significantly reduced the weight of tumors in the abdominal cavity and the number of mesenteric dissemination tumor nodules of mice (Fig. 7B,C), suggesting that IL-33 combined with p38 inhibitor has synergistic anti-tumor effect on abdominal dissemination tumors.

Then we compared the infiltration of macrophages subsets in different experimental groups by flow cytometry. The proportion of CD163$^+$ ($P = 0.0110$) (Fig. 7D) and CD206$^+$ ($P = 0.0033$) (Appendix Fig. S11A) macrophages decreased, and the proportion of CD80$^+$ ($P = 0.0202$) (Fig. 7E), CD86$^+$ ($P = 0.0048$) (Appendix Fig. S11B) and MHC-II$^+$ ($P = 0.0273$) (Appendix Fig. S11C) macrophages increased in the combined treatment group compared with the IL-33 monotherapy group. The aforementioned results indicated that the p38 signaling pathway could influence macrophage polarization and inhibiting p38 signaling pathway could reprogram macrophages from M2 to M1 phenotype. IL-33 combined with p38 signaling pathway inhibitor synergistically enhances the anti-tumor response of abdominal dissemination in GC.

Taken together, these data indicate that IL-33 can drive the recruitment and activation of immunologic effector cells and promote the expression of anti-tumor inflammatory cytokines, which reprogram the immunosuppressive TIME into immunoactivated state and inhibit tumor progression and metastasis. However, IL-33 also can induce M2 polarization through p38-GATA3 signaling pathway, which fosters tumor progression and resolution of inflammation. Thus, the combination of IL-33 and anti-CSF1R or p38 inhibitor could avoid the negative immune regulation of IL-33-induced M2 polarization, which has synergistic anti-tumor effect (Fig. 7F). P38 inhibitor can effectively inhibit the p38-GATA3 signaling pathway, reduce the expression of transcription factor GATA3, and inhibit the M2 polarization of macrophages.

## Discussion

PM of GC is the main cause of death in patients with advanced GC. Once PM is diagnosed, the median survival time is only 4 months, compared with 14 months for GC without PM (Chen et al, 2021). Emerging data highlight the role of the peritoneal TIME as the key driver of metastatic progression and treatment resistance (Deepak et al, 2020). Novel therapeutic approaches targeting the PM-associated TIME may be vital in preventing and reversing peritoneal cancer progression. In this study, we described that the cytokine IL-33 could extend survival in patients with GC and drive an activated immune microenvironment. Local intraperitoneal administration of IL-33 drove the recruitment and activation of immunologic effector cells and promoted the expression of pro-inflammatory cytokines. These changes in the TIME could promote a better prognosis for mice with GC.

IL-33, a member of the IL-1 family, can be rapidly released and binds to its cognate receptor ST2 in response to wound healing and inflammation (Jiang et al, 2021). In recent years, IL-33 has been described as a potent initiator of type 2 immune responses, acute local inflammation, and tissue repair, promoting cancer development and remodeling the TIME by increasing the number of immune-suppressive cells (Liew et al, 2016). Moreover, IL-33 can also promote the infiltration of immunologic effector cells, such as NK cells, CD8$^+$T cells, and TH1 immune cells and plays an anti-tumor role in several cancers (Fournié and Poupot, 2018; Sekiya et al, 2019). IL-33 can activate immunologic effector cells and induce innate immunity, suggesting that IL-33 has a unique approach and immunotherapeutic potential for treating peritoneal tumors. However, these dual effects of IL-33 in cancer imply that IL-33-based cancer immunotherapies should be considered with caution. Therefore, a more comprehensive exploration and deeper

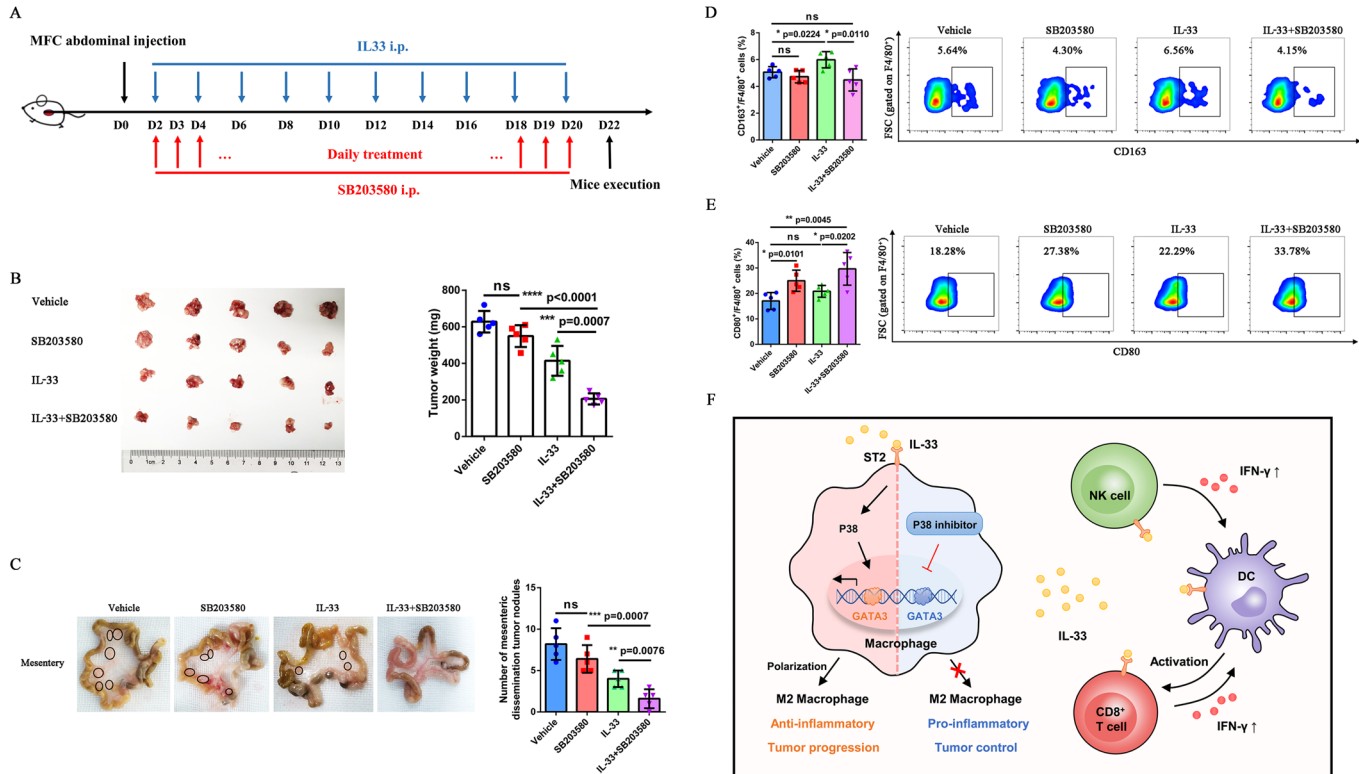

**Figure 7. IL-33 and p38 signaling pathway inhibitor synergistically induced antitumor response in vivo.**

(A) 615-line mice ($n = 5$ per group) developing abdominal dissemination upon intraperitoneal injection of MFC were treated with vehicle, SB203580, IL-33, or IL-33 + SB203580. Vehicle-treated mice served as controls. (B) Abdominal dissemination tumors and the tumor weight in MFC-challenged 615 mice treated with vehicle, SB203580, IL-33, or IL-33 + SB203580 ($n = 5$ biological replicates). (C) Mesenteric dissemination tumors and the number of mesenteric dissemination tumor nodules in MFC-challenged 615 mice treated with vehicle, SB203580, IL-33, or IL-33 + SB203580 ($n = 5$ biological replicates). (D) Proportions of CD163+/F4/80+ macrophages in abdominal tumors were determined by flow cytometry ($n = 5$ biological replicates). (E) Proportions of CD80+/F4/80+ macrophages in abdominal tumors were determined by flow cytometry ($n = 5$ biological replicates). (F) Local administration of IL-33 induced the reprogramming of the immunosuppressive TIME by enhancing the function of cytotoxic lymphocytes such as CD8+ T cells and NK cells, and promoted the formation of celiac inflammatory milieu by increasing the expression of inflammatory cytokines. On the contrary, IL-33 also promoted M2 polarization of macrophages and resulted in the resolution of inflammation through the p38-GATA3 signaling pathway simultaneously, which fostered tumor progression and metastasis. IL-33 combined with p38 inhibitor synergistically induced antitumor response in vivo. Data information: The data with error bars are shown as mean ± SD. ns, not significant; *$P < 0.05$, **$P < 0.01$, ***$P < 0.001$, ****$P < 0.0001$ by two-tailed unpaired-sample Student $t$ test. Also, see Appendix Fig. S11. Source data are available online for this figure.

understanding of IL-33 are required. In our GC tumor model, the increase of IL-33 levels locally resulted in the recruitment of CD8+T cells, NK cells, DCs, and macrophages and increased the expression of IFN-γ in CD8+ T cells and NK cells, indicating the activation of immunologic effector cells. This study suggested that the local administration of IL-33 could induce the celiac inflammatory environment, activate immunologic effector cells, promote the expression of antitumor cytokines, and reprogram the immunosuppressive microenvironment to be immunogenic. The advantage of an allergic-like antitumor response is that the effectors are located in the innate immune compartment and do not require specific initiation or generation of CTL immunity, suggesting a unique approach for treating peritoneal tumors using IL-33.

However, the immunoregulatory effects of IL-33 on immunocytes and the TIME are diverse and complex. IL-33 can not only promote the secretion of inflammatory cytokines such as IFN-γ and TNF-α and induce the formation of an inflammatory milieu, but also promote the recruitment of TAMs and M2 polarization of

macrophages to a certain extent, which has an immunosuppressive effect. This characteristic of IL-33 helps maintain the immune microenvironment in a relatively balanced and stable state, avoiding inflammatory injury of tissues and organs; this is also one of the reasons why some studies believe that IL-33 may promote tumor progression (Shen et al, 2018). In this study, we found that the proportion of M2 macrophages in peritoneal tumors of GC increased after IL-33 administration. The RNA-seq of mouse tumors also showed the upregulation of macrophage-related markers and activation of macrophage-related pathways in the IL-33 treatment group, which was consistent with the results of bioinformatic analysis and previously published studies (Faas et al, 2021; Okuzumi et al, 2021). These studies suggested that the cytokine IL-33 could recruit macrophages and regulate the M2 polarization of macrophages.

Peritoneal macrophages are the major cell type of peritoneal cells and are involved in many aspects of innate and acquired immunities; they can suppress antitumor immunity and promote tumor progression (Liu et al, 2018). Accumulating studies have

shown that peritoneal macrophages strongly express CD206, which is a characteristic phenotype of M2-polarized macrophages (Liu et al, 2015; Umemura et al, 2008). Functionally, peritoneal macrophages participate in various immune and inflammatory responses, which are related to the pathogenic processes of various inflammatory diseases and abdominal cancers. An attractive hypothesis regarding the mechanisms of TAM-mediated inflammation in cancer is that unidentified modifications might occur in regulating pro-inflammatory pathways during cancer evolution (Christofides et al, 2022). The properties of TAMs gradually change and lose their efficacy during the immune equilibrium phase. Only acute engagement of pro-immune inflammatory pathways during immune escape can override the continuing pro-cancer inflammation and trigger antitumor immunity, supporting the aforementioned hypothesis (Zhivaki et al, 2020). Macrophage intervention combined with IL-33 can induce strong and long-lasting protection against cancer and eradicate various tumors under such conditions. It is necessary to explore the unknown mechanism of peritoneal macrophages promoting tumor progression and metastasis, and understand the detailed biological characteristics and related molecular spectrum of the TIME to further control the development of peritoneal tumors.

We evaluated the stimulation of IL-33 on RAW264.7 cells, BMDM, THP-1 cells, and macrophages from PBMCs in vitro, and found that IL-33 could induce the polarization of M1 and M2 simultaneously, which was not completely consistent with the in vivo results. We believed that the TIME in vivo, especially in peritoneal tumors, was in a highly immunosuppressive state. Multiple immunosuppressive cytokines, chemokines, and immune checkpoints were highly expressed, leading to the failure of M1 polarization induced by IL-33 in vivo. This study examined both in vitro and in vivo models to obtain more comprehensive data.

Based on the essential role of M2 macrophages in promoting tumor growth and metastasis, we further reduced the infiltration of M2 macrophages in the TIME by eliminating TAMs or inhibiting M2 polarization. CSF1R antibody can reduce the number of TAMs and reprogram the polarization of macrophages. CSF1R targeting rarely eliminates tumors, but the therapeutic efficacy can be improved when combined with other agents (Engblom et al, 2016). In this study, IL-33 was combined with CSF1R antibody to activate the TME and reduce the infiltration of M2 macrophages, thus avoiding the negative immune regulation of IL-33-induced M2 polarization. The results of in vivo experiments also showed that the CSF1R antibody administered 10 days before tumor inoculation could eliminate macrophages earlier and more thoroughly compared with treatment after tumor inoculation, inducing a better antitumor effect, which indicated the importance of macrophages in the abdominal dissemination of GC.

Furthermore, we conducted additional investigations to elucidate the underlying mechanism of IL-33-induced polarization of macrophages. Our findings revealed that three classical signaling pathways, NF-κB p65, p44/42 MAPK, and p38, could be activated by IL-33. The activation of NF-κB p65 and p44/42 MAPK signaling pathways contributes to M1 polarization. On the contrary, activating the p38 signaling pathway promotes M2 polarization. When the p38 signaling pathway is inhibited, the M2 polarization tendency of macrophages is weakened. Previous studies demonstrated that the transcription factor GATA3 of the p38 signaling

pathway was closely correlated with the decrease in inflammation and tissue repair of macrophages (Jia et al, 2022; Lin et al, 2019). We found that IL-33 could effectively activate the p38-GATA3 signaling pathway and promote the expression of GATA3, proving that IL-33 could induce M2 polarization by activating the p38-GATA3 signaling pathway. The results of combination therapy showed that IL-33 combined with p38 inhibitor could reprogram macrophages from the M2 phenotype to the M1 phenotype and had a synergistic antitumor effect, further confirming the mechanism in in vitro experiments. Combined with the characteristic of M2 polarization following the activation of the p38-GATA3 signaling pathway, we believed that the transcription factor GATA3 was a key factor in IL-33-regulated M2 polarization.

This study had a limitation in terms of using a homogeneous mouse model, which allowed us to dissect IL-33-induced TIME reprogramming of the abdominal dissemination of GC. Studies are needed to better address the significance of these findings in other mouse models or humans.

## Conclusions

We showed that the intraperitoneal administration of IL-33 was a promising strategy for treating peritoneally confined metastatic cancers. The survival advantage conferred by this therapy depended on the activation of immunocytes, the formation of a celiac inflammatory environment, and the reprogramming of an immunosuppressive TIME. This study additionally focused on the dual role of IL-33 in antitumor activity and identified a critical signaling pathway, p38-GATA3, which promoted the M2 polarization of macrophages and simultaneously fostered tumor progression and resolution of inflammation. This study provided a theoretical basis and novel approach for the antitumor strategy of IL-33 combined with anti-CSF1R or M2 polarization inhibitors. It is essential to investigate further the local delivery of this unique cytokine for tumor therapy.

## Methods

### Data source and processing

Transcriptomic and matched clinical data of patients with GC in TCGA-STAD, ACRG/GSE66229 (Data ref: Oh et al, 2015; Oh et al, 2018), and GSE162214 (Data ref: Tanaka, 2021) are available from the TCGA and GEO databases. The relationship between *IL-33* mRNA expression and prognosis was evaluated using the Kaplan–Meier Plotter (https://kmplot.com/analysis/index.php?p=service&cancer=gastric) (Gene symbol: 209821_at), an online database containing gene expression data and survival information of patients with GC (Győrffy et al, 2013). The TME score data were calculated using the xCell algorithm (Aran et al, 2017).

A total of 170 patients histologically diagnosed with GC from August 2017 to January 2019 in the Nanjing Drum Tower Hospital were included in this study. Kaplan–Meier survival curve was used to compare the difference in OS between patients with high and low IL-33 expression, with H-score = 4 as the discriminant critical value. The hazard ratio with a 95% confidence interval and log-rank $P$ value were calculated.

## Cell lines and cell culture

Human GC cell lines AGS (ATCC CRL-1739), SUN601 (KANGBAI CBP60507), NUGC4 (KANGBAI CBP74135), MKN45 (KANG-BAICBP60488), MGC803 (KANGBAI CBP60485), HGC27 (KANG-BAI CBP60480), and KATOIII (KANGBAI CBP60483); human acute monocytic leukemia cell line THP-1 (KANGBAI CBPB0007); and murine GC cell line MFC (KANGBAI CBP60882) were cultured in Roswell Park Memorial Institute (RPMI)-1640 medium (Gibco). Murine macrophage cell line RAW264.7 (ATCC SC-6003) was cultured in Dulbecco's modified Eagle's medium (DMEM, Hyclone). All culture mediums were supplemented with 10% fetal bovine serum (FBS, Biochrom), 100 U/mL penicillin, and 100 μg/mL streptomycin (Gibco). The mycoplasma contamination was examined routinely using a PCR mycoplasma detection kit every month.

## Antibodies and reagents

Commercially available antibodies and reagents used in this study were listed in Appendix Tables S2, S3.

## Lentiviral production and creation of stable cell lines

Lentiviruses treated with luciferase report vectors were designed by Cyagen (Suzhou, China) and used to infect MFC cells for 48 h. Further, 2 μg/mL puromycin was used for screening stable luciferase-transduced MFC cells (MFC-Luc).

## Construction and in vitro transfection of GATA3-shRNA

GATA3-shRNA and scramble-shRNA were synthesized by Shanghai Genechem Co., Ltd., (Shanghai, China). RAW264.7 cells were transfected with GATA3-shRNA using lipofectamine 2000 as previously described.(Hernandez-Alejandro et al, 2012) The scramble-shRNA transfected cells were used as controls.

## Culture and differentiation of BMDMs

BMDMs were collected from 615-line mice aseptically by rinsing the bilateral femurs and tibias of euthanized mice with PBS and treating them with erythrocyte lysate. Purified BMDMs were cultured in RPMI-1640 with 10% FBS and 50 ng/mL M-CSF. More than 90% of BMDMs were F4/80$^+$ detected by flow cytometry and collected for macrophage stimulation on day 5.

## In vitro mouse macrophage polarization and treatment

RAW264.7 cells and BMDMs of 615-line mice were polarized into classically activated macrophages (M1) with 100 ng/mL LPS, or alternatively activated macrophages (M2) with mouse IL-4 (25 ng/mL) and IL-10 (25 ng/mL) recombinant protein for 48 h. The THP-1 monocytes were differentiated into macrophages using 10 ng/mL phorbol-12-myristate-13-acetate (PMA) for 24 h. The THP-1 macrophages were subsequently stimulated using human IFN-γ (50 ng/mL) and LPS (50 ng/mL) for 48 h to the M1 phenotype, or stimulated using human IL-4 (25 ng/mL) and IL-13 (25 ng/mL) for 48 h to M2 phenotype. The untreated RAW264.7 cells and PMA-stimulated THP-1 cells were used as M0 phenotype.

## Isolation of human monocytes

PBMCs (collected from patients with GC after informed consent was obtained) were obtained by Ficoll density centrifugation (Zhou et al, 2021). The monocyte population was enriched by the negative selection of unlabeled target cells using a pan monocyte isolation kit following the manufacturer´s protocol. The monocytes were cultured for 7 days with no further exogenous agent was added to allow spontaneous differentiation of monocytes into resting (M0) macrophages.

## Cytotoxicity assay

FITC-CFSE labeling of SNU601 cells was performed in a 1:1 ratio with a final concentration of CFSE at 2.5 μg/mL, allowed to incubate at room temperature for 8 min while protected from light (Ganesan et al, 2022). SNU601 cells were co-cultured with PBMCs for 12 h after stimulating PBMCs with or without human recombinant IL-33 (50 ng/mL) for 48 h. The cell suspension was incubated in the dark for 15 min after adding propidium iodide with a final concentration of 0.5 μg/mL.

## RNA extraction and qRT-PCR

Total RNA was extracted from cultured human GC and THP-1 cells using the TRIzol reagent following the manufacturer's protocol. cDNA was synthesized using the amfiRivert cDNA Synthesis Platinum Master Mix. Each cDNA sample was amplified using Power SYBR Green PCR Master Mix on the ABI QuantStudio 7 Flex real-time PCR system (Applied Biosystems). GAPDH was used as an endogenous control to normalize each sample. The experiment was performed in triplicate with three independent experiments. The fold changes in the mRNA expression of these genes were calculated using the $2^{-\Delta\Delta Ct}$ method for relative quantification. The primers are listed in Appendix Table S4.

## Hematoxylin–eosin staining and IHC assay

Mouse organ tissues and human GC tissue samples were prepared as 5 μM FFPE sections. Paraffin-embedded mouse organ tissue sections were stained with hematoxylin and eosin (H&E). Immunohistochemistry was performed on the GC tissue sections using IL-33 monoclonal antibody and biotinylated goat anti-rabbit serum immunoglobulin G (IgG) antibody.

The estimation of IL-33 staining in tissue sections was performed by the proportion of positive cells and staining intensity (Yue et al, 2020). The proportion of positive cells in the sections was categorized into five grades: 0, <5%; 1, 5–25%; 2, 26–50%; 3, 51–75%; and 4, >75%. The staining intensity was categorized into four grades: 0, no staining; 1, weakly positive; 2, moderately positive; and 3, strongly positive. The H-score (ranging from 0 to 12) was calculated by multiplying the proportion of positive cells by the intensity score. The staining results were assessed independently by two experienced pathologists. The optimal cut-off value was determined using the X-tile software for prognostic analysis. H-score ≤ 4 indicated low IL-33 expression, and H-score > 4 indicated high expression.

## Flow cytometry analysis and intracellular staining

Tumor tissues and spleens from GC models in vivo and macrophages in vitro were harvested and mechanically dissociated into single-cell suspensions to evaluate immune cell markers. The cells were stimulated with Leukocyte Activation Cocktail and treated with a fixation/permeabilization solution kit to assess the expression of intracellular cytokines. Then, the cells were incubated with antibodies to assess the immune cell surface markers. The acquisition was performed using a BD Accuri C6 Plus Flow Cytometer (BD Bioscience), and the data were analyzed using FlowJo software (10.4, Tree Star).

## Cytometric bead array

The levels of IFN-γ, IL-2, IL-5, TNF-α, IL-6, IL-4, IL-10, and IL-13 in the supernatant of ascites and cultured RAW264.7 cells were determined using the multi-analyte flow assay kit following the manufacturer's protocols.

## Western blotting

RAW264.7 cells were starved in 2% FBS-DMEM overnight, followed by pre-treatment with selective inhibitors against NF-κB (JSH-23, 20 μM), MEK1/2 (U0126, 10 μM), or p38 (SB203580, 20 μM) for 1 h before stimulation with IL-33 (50 ng/mL) for 15 min as previously described (Staudt et al, 2013; Yang et al, 2016; Zhang et al, 2018; Zhou et al, 2018). Further, 10 μg of total protein from each sample was subjected to 10% SDS-polyacrylamide gel electrophoresis and transferred onto polyvinylidene difluoride membranes. Then, the membranes were incubated with primary antibodies and secondary antibodies conjugated with HRP. The protein bands were detected using Immobilon Western HRP Substrate (Merck, WBKLS0050).

## Immunofluorescence staining

RAW264.7 cells were pretreated with or without selective inhibitors JSH-23 (20 μM), U0126 (10 μM), and SB203580 (20 μM) for 1 h. The cells were fixed and permeabilized after stimulation with or without mouse IL-33 recombinant protein for 36 h. The cells were incubated with phalloidin–TRITC to detect the cytoskeleton and incubated with primary antibodies and fluorescence-labeled secondary antibodies to detect GATA3. The nuclei were stained with DAPI, and the images were acquired using a fluorescent microscope (Leica).

The tumor tissue sections of MFC-challenged mice were prepared as described in the IHC analysis. The tumor tissue sections were stained with the anti-CD86 antibody and anti-CD163 antibody at 4 °C overnight. Then, the sections were incubated with secondary antibodies, protected from light. The nuclei were stained with DAPI, and the images were acquired using a fluorescent microscope (Leica).

## Mice

615-line mice (catalog number: M0026), the MFC GC cell line syngeneic murine model, were purchased from Cavens Biotechnology Co., Ltd. (Nanjing, China) and housed in a specific pathogen-free facility with a relative humidity of 55% ± 5%, room temperature of 22 °C ± 2 °C, and 12-h/12-h light/dark cycle. The mice had free access to food and water during the whole experiment. They acclimatized to the laboratory for at least 1 week before initiating the study. All animal experiments were performed by blinded researchers. Six-week-old female or male mice were assorted in a randomized manner.

## Syngeneic mouse tumor models

Sex-matched six-week-old 615-line mice were used for this study. For the abdominal dissemination tumor model, $5 \times 10^5$ MFC cells per animal were injected intraperitoneally (i.p.) into mice ($n = 4$ or 5 per group). For the subcutaneous tumor model, $1 \times 10^6$ MFC cells per animal were injected subcutaneously (s.c.) into mice ($n = 5$ per group). During treatment, the mice were injected i.p. with 20 μg/kg IL-33 every other day, 400 μg anti-CSF1R per mouse every 3 days, and 5 mg/kg SB203580 every day as previously described (Barkal et al, 2019; Dalmas et al, 2017; Leelahavanichkul et al, 2014).

The mouse body weight and tumor size were monitored every other day during the experiment. The tumor size was calculated using the following formula: length × width$^2$ × 0.5. The mice were euthanized after the treatment. For safety studies, one mouse from each group was randomly selected, and the main organs were collected for H&E staining.

## In vivo macrophage depletion treatment study

In this study, 615-line mice were treated with anti-CSF1R as proposed by Barkal et al (Barkal et al, 2019). The mice were treated from day 2 or pretreated for 10 days with 400 μg anti-CSF1R per mouse every 3 days.

## In vivo bioluminescence imaging (BLI)

Mice implanted with MFC-Luc received an intraperitoneal injection of D-luciferin stock solution (150 mg/kg) to produce bioluminescence. Ventral bioluminescent images were taken every 4 days. Anesthesia was maintained during the entire imaging process using a nose cone isoflurane-oxygen delivery device in the light-tight chamber. Photon signaling intensity was quantitatively recorded to obtain the dynamic changes in bioluminescence in vivo by IVIS Lumina imaging system (Xenogen).

## Transcriptome sequencing (RNA-seq) analysis

RNA-seq and subsequent bioinformatics analysis were performed by BerryGenomics (Beijing, China). Tumor samples from the vehicle and IL-33 treatment groups ($n = 8$ per group) with sufficiently high-quality RNA were included for RNA-seq.

## Statistical analysis

R version 3.6.3 via RStudio software, GraphPad Prism 7.0 (GraphPad Software, CA, USA), and SPSS Statistics version 20.0 (IBM) were used for all statistical analyses. We employed a random number table to perform randomization. All animal experiments in this study were performed and analyzed in a blinded manner. Treatment groups were assigned in a randomized fashion. Every

**The Paper Explained**

**Problem**

Peritoneal metastasis is the most common form of recurrence and metastasis for gastric cancer and the main cause of death in patients. The suppressive tumor immune microenvironment (TIME) limits the effects of immunotherapy.

**Results**

The intraperitoneal administration of IL-33 induced a celiac inflammatory milieu and inhibited metastasis progression. IL-33 could also activate downstream signaling pathways, including NF-κB, p38, and p44/42, and induce M2 polarization of macrophages by activating the p38-GATA-binding protein 3 signaling pathway. IL-33 combined with an anti-CSF1R or p38 inhibitor to regulate TAMs had a synergistic antitumor effect in the abdominal dissemination model.

**Impact**

These results provide a promising novel approach for treating metastatic peritoneal malignancies, which, if combined with TAM reprogramming to reshape the TIME, could achieve a better therapeutic outcome. Further investigation is warranted to test whether local delivery of this unique cytokine would be useful for tumor therapy in human patients.

mouse was assigned a temporary random number within the weight range. The mice were given their permanent numerical designation in the cages after they were randomly categorized into each group. For each group, a cage was selected randomly from the pool of all cages. All data were collected and analyzed by two blinded observers. No statistical methods were used to predetermine sample sizes. No analyzed samples were omitted from the report. The samples were omitted from the analysis if insufficient material was available. The data were represented as five biological replicates in animal experiments and three or five biological replicates in cell experiments and expressed as mean ± standard deviation. A $P$ value < 0.05 indicated a statistically significant difference. The two-tailed unpaired-sample Student $t$ test, log-rank test, or Wilcoxon rank-sum test was used for comparing between two groups. The Spearman correlation coefficient was determined using linear regression analysis (ns, not significant; $*P < 0.05$, $**P < 0.01$ and $***P < 0.001$, $****P < 0.0001$).

## Ethics approval and consent to participate

The tumor tissue sections and peripheral blood collection procedures were carried out in accordance with the guidelines verified and approved by the Ethics Committee of Nanjing University Medical School Affiliated Drum Tower Hospital (Approved number: 2021-324-01), and the experiments conformed to the principles set out in the WMA Declaration of Helsinki and the Department of Health and Human Services Belmont Report. All subjects signed an informed consent for scientific research statement. Because the patients in the public databases could not be identified, the analysis and reporting of the data in our study were exempt from review by the Ethics Committee of Nanjing Drum Tower Hospital. The requirement for written informed consent to participate was waived. All animal procedures were conducted in accordance with the NIH Guide for Care and Use of Laboratory Animals and approved by the Institutional Animal Care and Use Committee of Nanjing Drum Tower Hospital (Approved number: 20191013).

## Data availability

The clinicopathological characteristics of gastric cancer patients in the paper are detailed in Dataset EV1. RNA-seq data generated for this study have been deposited in the Gene Expression Omnibus repository with the accession number GSE235526.

## Peer review information

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

## Acknowledgements

This work was supported by grants from National Key R&D Program of China (grant number 2023YFC2506400), the Fundamental Research Funds for the Central Universities (0214-14380506), the Incubation Foundation of Shandong

Provincial Hospital (2022FY005), and the Natural Science Foundation of Shandong Province (ZR2023QH299).

## Author contributions

**Keying Che**: Conceptualization; Data curation; Formal analysis; Supervision; Funding acquisition; Validation; Investigation; Methodology; Writing—original draft; Writing—review and editing. **Yuting Luo**: Formal analysis; Investigation; Writing—review and editing. **Xueru Song**: Software; Validation; Visualization. **Zhe Yang**: Software; Formal analysis; Writing—review and editing. **Hanbing Wang**: Data curation; Software; Formal analysis. **Tao Shi**: Resources; Data curation; Software; Formal analysis; Writing—original draft. **Yue Wang**: Investigation; Methodology. **Xuan Wang**: Resources; Investigation. **Hongyan Wu**: Resources; Investigation. **Lixia Yu**: Resources; Supervision. **Baorui Liu**: Resources; Supervision. **Jia Wei**: Conceptualization; Supervision; Funding acquisition; Validation; Methodology; Writing—original draft; Project administration; Writing—review and editing.

## Disclosure and competing interests statement

The authors declare no competing interests.

