## [Peer Review File · EMBO Molecular Medicine]

Macrophages reprogramming improves immunotherapy of IL-33 in peritoneal metastasis of gastric cancer

Keying Che, Yuting Luo, Xueru Song, Zhe Yang, Hanbing Wang, Tao Shi, Yue Wang, Xuan Wang, Hongyan Wu, Lixia Yu, Baorui Liu, and Jia Wei

DOI: [10.15252/emmm.202217321](https://doi.org/10.15252/emmm.202217321)

Corresponding author: Jia Wei (jiawei99@nju.edu.cn)

Review Timeline:

Submission Date:	18th Dec 22
Editorial Decision:	7th Apr 23
Revision Received:	27th Jun 23
Editorial Decision:	5th Sep 23
Revision Received:	24th Sep 23
Accepted:	20th Nov 23

Editor: Kelly Anderson

Transaction Report:

7th Apr 2023

Dear Dr. Wei,

Thank you for the submission of your manuscript to EMBO Molecular Medicine. We have now received feedback from the reviewers who agreed to evaluate your manuscript. As you will see from the reports below, the referees acknowledge the interest of the study and are overall supporting consideration of an appropriate revision.

Addressing the reviewers' concerns in full will be necessary for further consideration of the manuscript in our journal, and acceptance of the manuscript will entail a second round of review. EMBO Molecular Medicine encourages a single round of revision only and therefore, acceptance or rejection of the manuscript will depend on the completeness of your responses included in the next, final version of the manuscript. For this reason, and to save you from any frustrations in the end, I would strongly advise against returning an incomplete revision. It would be good to discuss your plan to address the referee concerns and I am available to do so in the coming weeks by email or zoom.

Revised manuscripts should be submitted within three months of a request for revision; they will otherwise be treated as new submissions, except under exceptional circumstances in which a short extension is obtained from the editor. Also, the length of the revised manuscript may not exceed 60,000 characters (including spaces) and, including figures, the paper must ultimately fit onto optimally ten pages of the journal. You may consider including any peripheral data (but not methods in their entirety) in the form of Supplementary information.

I look forward to seeing a revised form of your manuscript as soon as possible.

Yours sincerely,

Kelly

Kelly M Anderson, PhD
Scientific Editor
EMBO Molecular Medicine

We require:

- 1) A .docx formatted version of the manuscript text (including legends for main figures, EV figures and tables). Please make sure that the changes are highlighted to be clearly visible.
- 2) Individual production quality figure files as .eps, .tif, .jpg (one file per figure). For guidance, download the 'Figure Guide PDF': (<https://www.embopress.org/page/journal/17574684/authorguide#figureformat>).
- 3) A .docx formatted letter INCLUDING the reviewers' reports and your detailed point-by-point responses to their comments. As part of the EMBO Press transparent editorial process, the point-by-point response is part of the Review Process File (RPF), which will be published alongside your paper.
- 4) A complete author checklist, which you can download from our author guidelines (<https://www.embopress.org/page/journal/17574684/authorguide#submissionofrevisions>). Please insert information in the checklist that is also reflected in the manuscript. The completed author checklist will also be part of the RPF.
- 5) Please note that all corresponding authors are required to supply an ORCID ID for their name upon submission of a revised manuscript.
- 6) It is mandatory to include a 'Data Availability' section after the Materials and Methods. Before submitting your revision, primary datasets produced in this study need to be deposited in an appropriate public database, and the accession numbers and database listed under 'Data Availability'. Please remember to provide a reviewer password if the datasets are not yet public (see <https://www.embopress.org/page/journal/17574684/authorguide#dataavailability>).

.

13) Author contributions: You will be asked to provide CRediT (Contributor Role Taxonomy) terms in the submission system. These replace a narrative author contribution section in the manuscript.

14) A Conflict of Interest statement should be provided in the main text.

15) Every published paper now includes a 'Synopsis' to further enhance discoverability. Synopses are displayed on the journal webpage and are freely accessible to all readers. They include a short stand first (maximum of 300 characters, including space) as well as 2-5 one-sentences bullet points that summarizes the paper. Please write the bullet points to summarize the key NEW

findings. They should be designed to be complementary to the abstract - i.e. not repeat the same text. We encourage inclusion of key acronyms and quantitative information (maximum of 30 words / bullet point). Please use the passive voice. Please attach these in a separate file or send them by email, we will incorporate them accordingly.

Please note: When submitting your revision you will be prompted to enter your funding and payment information. This will allow Wiley to send you a quote for the article processing charge (APC) in case of acceptance. This quote takes into account any reduction or fee waivers that you may be eligible for. Authors do not need to pay any fees before their manuscript is accepted and transferred to the publisher.

EMBO Press participates in many Publish and Read agreements that allow authors to publish Open Access with reduced/no publication charges. Check your eligibility: <https://authorservices.wiley.com/author-resources/Journal-Authors/open-access/affiliation-policies-payments/index.html>

***** Reviewer's comments *****

Referee #1 (Comments on Novelty/Model System for Author):

-

Referee #1 (Remarks for Author):

The manuscript by Che et al was to determine the potential therapeutic role of IL-33 in gastric cancer. Gastric cancer is still a major cancer nowadays, despite extensive clinical and basic research over the last decades. The current study demonstrated that IL-33, indeed, contributes to inhibit the development of gastric cancer in vivo and in vitro, via regulating infiltrating tumour associate macrophages.

1. Any expression of MCSF-1R in GC?
2. What is the half-life of IL-33 and anti MCSF-1R
3. The correlation of M1 vs M2 in GC patients, and the PBMC from GC patients in response to IL-33 stimulation in vitro, and possible cytotoxicity to GC cells in vitro? Rather than determine macrophage cell line in vitro.
4. p38-GATA3 signaling pathway in macrophages from GC patients in vivo and/or in vitro
5. ... IL-33 also can induce M2 polarization through p38-GATA3 signaling pathway for killing of GC. It is well known that M1 macrophages are anti-cancer but M2 promote cancer. Please explain. Or I am very much confused by the data presented.
6. The discussion was far too long, quite a large portion of the discussion is redundant
7. The manuscript would be benefit with language editing.

Referee #3 (Remarks for Author):

Figure 1D: It is easy to misunderstand two images to be one normal and the other tumor tissue. One representative image should be enough. It will be better to quantify the IL33 expression in normal and tumor tissue areas.

Figure 2, How many biological replicates of mice experiments have the authors done? How many mice per group? It seems there are 6 data points in Figure 2D-L, while only 5 for the rest.

Please indicate the inhibitor concentration used in the current study.

Figure 6D, please provide statistical data. And it will be useful to run WB for GATA3 expression.

The current data could not support the statement "IL-33 activates the p38-GATA3 signaling pathway to induce M2 polarization of macrophages", because there is no evidence of a relationship between GATA3 and macrophage polarization unless the author could show the inhibition of GATA3 can also skew the macrophage phenotype.

Figure 5 and 7, It will be better to provide a schematic illustration similar to Figure 2A.

Figure 7D, what about other M1/M2 markers (CD86, MHCII, CD206)?

Please provide scientific support for the dose/duration of IL-33, anti-CSF1R and SB203580.

Referee #1

The manuscript by Che et al was to determine the potential therapeutic role of IL-33 in gastric cancer. Gastric cancer is still a major cancer nowadays, despite extensive clinical and basic research over the last decades. The current study demonstrated that IL-33, indeed, contributes to inhibit the development of gastric cancer *in vivo* and *in vitro*, via regulating infiltrating tumour associate macrophages.

Thank you. We appreciate the time and patience you put into our work. We tried our best to improve the manuscript and made some changes in the manuscript. We have attempted, where possible, to amend the manuscript based on the comments. Please find below the responses to the different comments.

1. Any expression of MCSF-1R in GC?

Thank you. Okugawa et al. demonstrated that MCSF-1R is highly expressed in cancer cells compared with normal mucosa in GC tissues by immunohistochemical analysis. The reference is as follows:

Okugawa Y, Toiyama Y, Ichikawa T, et al. Colony-stimulating factor-1 and colony-stimulating factor-1 receptor co-expression is associated with disease progression in gastric cancer. *Int J Oncol.* 2018 Aug;53(2):737-749. PMID: 29767252.

2. What is the half-life of IL-33 and anti MCSF-1R?

Thank you for your comment. Previous studies showed that the half-life of IL-33 was 1.4 h (95% CI 1.2–1.6) in patients and the half-life of anti-CSF1R antibody (LY3022855) was more than 30 h in patients with advanced solid tumors. The above data can be used as a reference for research. The references are as follows:

Sundnes O, Ottestad W, Schjalm C, et al. Rapid systemic surge of IL-33 after severe human trauma: a prospective observational study. *Mol Med.* 2021 Mar 26;27(1):29. PMID: 33771098.

Dowlati A, Harvey RD, Carvajal RD, et al. LY3022855, an anti-colony stimulating factor-1 receptor (CSF-1R) monoclonal antibody, in patients with advanced solid tumors refractory to standard therapy: phase 1 dose-escalation trial. *Invest New Drugs*. 2021 Aug;39(4):1057-1071. PMID: 33624233.

Due to the difference in experimental conditions and reagent types, the dose and duration of IL-33 and anti-CSF1R antibody also varies to some degree. We designed our experiments with reference to previous similar studies. *In vivo* experiments, the dose and duration of IL-33 and anti-CSF1R antibody were used as previous described by Dalmas et al. in *Immunity* and Barkal et al. in *Nature* respectively. *In vitro* experiments, the dose of IL-33 was used as previous described by Yang et al. in *Nature Communication*. The references are as follows:

Dalmas E, Lehmann FM, Dror E, et al. Interleukin-33-Activated Islet-Resident Innate Lymphoid Cells Promote Insulin Secretion through Myeloid Cell Retinoic Acid Production. *Immunity*. 2017 Nov 21;47(5):928-942.e7. PMID: 29166590.

Barkal AA, Brewer RE, Markovic M, et al. CD24 signalling through macrophage Siglec-10 is a target for cancer immunotherapy. *Nature*. 2019 Aug;572(7769):392-396. PMID: 31367043.

Yang Y, Andersson P, Hosaka K, et al. The PDGF-BB-SOX7 axis-modulated IL-33 in pericytes and stromal cells promotes metastasis through tumour-associated macrophages. *Nat Commun*. 2016 May 6;7:11385. PMID: 27150562.

We have added and highlighted the explanations and references in the Materials and Methods section (p35 and p37 of revised manuscript).

Thank you again for your suggestion.

3. The correlation of M1 vs M2 in GC patients, and the PBMC from GC patients in response to IL-33 stimulation *in vitro*, and possible cytotoxicity to GC cells *in vitro*? Rather than determine macrophage cell line *in vitro*.

Thank you for this comment. PBMCs (collected from GC patients after informed consent was obtained) were obtained by Ficoll density centrifugation. Monocytes were isolated from PBMCs of GC patients and differentiated into resting (M0) macrophages. For macrophages isolated from the PBMCs of patients with GC, the expression of CD80 ($P = 0.0040$) and CD163 ($P = 0.0031$) were increased by IL-33 stimulation *in vitro* (Appendix Fig S5A). Next, we examined the effects of IL-33 on PBMC-mediated killing to GC cells. At low E:T ratios such as 1:1, IL-33 readily induced substantial apoptosis of SNU601 cells ($P = 0.0006$), suggesting that IL-33 is essential for PBMC-mediated killing to tumor cells (Appendix Fig S5B). Appendix Fig S5 is as follows.

We have added and highlighted relevant data and explanation in the Results section (p13 of revised manuscript).

4. p38-GATA3 signaling pathway in macrophages from GC patients *in vivo* and/or *in vitro*

Thank you for your comment. Monocytes were isolated from PBMCs of GC patients and differentiated into resting (M0) macrophages. We further evaluated the expression of GATA3, CD80, and CD163 in human macrophages to clarify the influence of p38-GATA3 signaling pathway-induced macrophage polarization. The expression of GATA3 ($P = 0.0041$; Appendix Fig S10A) and CD163 ($P = 0.0002$; Appendix Fig S10B) significantly decreased when macrophages were stimulated with IL-33 + SB203580 compared with IL-33 alone. On the contrary, the expression of CD80 ($P = 0.0384$) increased when the macrophages were treated with IL-33 + SB203580 compared with IL-33 alone (Appendix Fig S10C), suggesting that p38 inhibitor could effectively inhibit the expression of GATA3 and block the M2 polarization induced by p38-GATA3 signaling pathway. The relevant data was presented and highlighted in the Results section (p18 of revised manuscript). Appendix Fig S10 is as follows.

And we do agree with your comment. It is constructive and important to study p38-GATA3 signaling pathway in macrophages of GC patients *in vivo*. We have demonstrated the effect of p38-GATA3 signaling pathway on macrophages by mice model and we will certainly consider performing clinical trials on the basis of sufficient research and detailed information to verify the results for future work. Thank you very much for your kind suggestions again.

5. ... IL-33 also can induce M2 polarization through p38-GATA3 signaling pathway for killing of GC. It is well known that M1 macrophages are anti-cancer but M2 promote cancer. Please explain. Or I am very much confused by the data presented.

Thank you. Locally administration of IL-33 induces the reprogramming of immunosuppressive TME by enhancing the function of cytotoxic lymphocytes such as CD8⁺ T cells and NK cells, and promotes the formation of celiac inflammatory milieu by increasing the expression of inflammatory cytokines. On the contrary, IL-33 also can promote M2 polarization of macrophages and result in the resolution of inflammation through p38-GATA3 signaling pathway simultaneously, which fosters tumor

progression and metastasis. IL-33 combined with p38 inhibitor synergistically induces anti-tumor response *in vivo*.

In order to make this clear, we have changed the sentence “Moreover, IL-33 also can induce M2 polarization through p38-GATA3 signaling pathway, so the combination of IL-33 and anti-CSF1R or p38 inhibitor has synergistic anti-tumor effect (Fig 7E).” to “However, IL-33 also can induce M2 polarization through p38-GATA3 signaling pathway, which fosters tumor progression and resolution of inflammation. Thus, the combination of IL-33 and anti-CSF1R or p38 inhibitor could avoid the negative immune regulation of IL-33-induced M2 polarization, which has synergistic anti-tumor effect (Fig 7F).” in the tailender paragraph of Results section (p20 of revised manuscript). Thank you again for your comment.

6. The discussion was far too long, quite a large portion of the discussion is redundant.

Thank you for your advice. We have modified the discussion and removed redundant parts.

7. The manuscript would be benefit with language editing.

Thank you for your kind advice. To improve the readability of this manuscript, language editing has been performed by a native speaker.

Referee #3

1. Figure 1D: It is easy to misunderstand two images to be one normal and the other tumor tissue. One representative image should be enough. It will be better to quantify the IL33 expression in normal and tumor tissue areas.

Thanks so much for your kind suggestion on our manuscript. Only one representative image was kept in Fig 1D. We found 42 IHC sections with normal tissues. The expression of IL-33 in normal and tumor tissue areas was quantified by staining intensity, which was divided into four grades: 0, no staining; 1, weakly positive; 2, moderately positive; 3, strongly positive. The staining intensity of IL-33 in normal tissues was higher than that in tumor tissues ($P = 0.0476$). The data was added and highlighted in the Results section (p7 of revised manuscript). Thank you again for your suggestion.

2. Figure 2, How many biological replicates of mice experiments have the authors done? How many mice per group? It seems there are 6 data points in Figure 2D-L, while only 5 for the rest.

Thanks so much for your comment. We included 6 mice per group in the pre-experiment and 5 mice in the formal experiment. We had previously shown the flow cytometry results of the pre-experiment accidentally in Fig 2D-L. And we have now corrected the results to formal experimental results. The trend was same in the formal and pre-experiments. The flow cytometry results of pre-experiment and formal experiment are as follows.

We have revised and highlighted the relevant data in the manuscript (p9 of revised manuscript). And we will review the experimental data more carefully in the future. Thank you again for your constructive comment.

Pre-experiment

Formal experiment

3. Please indicate the inhibitor concentration used in the current study.

Thank you for your comment. The concentrations of inhibitors JSH-23 (20 μ M), U0126 (10 μ M) and SB203580(20 μ M) were supplemented and highlighted in the Materials and Methods section (p35 of revised manuscript). And we added references about inhibitor concentrations in the manuscript (references are as follows).

Staudt ND, Jo M, Hu J, et al. Myeloid cell receptor LRP1/CD91 regulates monocyte recruitment and angiogenesis in tumors. *Cancer Res.* 2013 Jul 1;73(13):3902-12. PMID: 23633492.

Zhang X, Qin J, Zou J, et al. Extracellular ADP facilitates monocyte recruitment in bacterial infection via ERK signaling. *Cell Mol Immunol.* 2018 Jan;15(1):58-73. PMID: 27867196.

Zhou MM, Zhang WY, Li RJ, et al. Anti-inflammatory activity of Khayandirobilide A from *Khaya senegalensis* via NF- κ B, AP-1 and p38 MAPK/Nrf2/HO-1 signaling pathways in lipopolysaccharide-stimulated RAW 264.7 and BV-2 cells. *Phytomedicine.* 2018 Mar 15;42:152-163. PMID: 29655681.

Thank you for your comment again.

4. (1) Figure 6D, please provide statistical data.

(2) And it will be useful to run WB for GATA3 expression.

(1) Thank you for your comments. The immunofluorescence staining for GATA3 in RAW264.7 cells was quantitatively analyzed. The immunofluorescence staining assay showed GATA3 level significantly increased in RAW264.7 cells stimulated with IL-33 ($P = 0.0009$). The expression of GATA3 also decreased in the IL-33 + SB203580 group compared with the IL-33-stimulated group ($P = 0.0077$) (Fig 6E). We have added and highlighted the data in Results section (p18 of revised manuscript).

(2) Thank you for the comment. GATA3 expression was evaluated by flow cytometry instead of WB. The expression of GATA3 was inhibited when the cells were stimulated with IL-33 + SB203580 compared with IL-33 alone ($P = 0.0021$) (Fig 6D). We have added and highlighted the data in Results section (p18 of revised manuscript). Thank you again for your suggestions.

5. The current data could not support the statement "IL-33 activates the p38-GATA3 signaling pathway to induce M2 polarization of macrophages", because there is no evidence of a relationship between GATA3 and macrophage polarization unless the author could show the inhibition of GATA3 can also skew the macrophage phenotype.

Thank you for your constructive comment. GATA3-shRNA was constructed to knock down the GATA3 expression and activity in RAW264.7 cells to verify the role of GATA3 in M2 polarization. The expression of CD163 significantly decreased in

GATA3-shRNA-transfected cells stimulated with IL-33 compared with that in the IL-33 + scramble-shRNA group ($P = 0.0001$) (Fig 6F). Above data accordingly suggest a critical role of GATA3 in macrophages that responds to an IL-33-induced reprogramming. We have added and highlighted the data in Results section (p18 of revised manuscript). Thank you for your kind suggestion.

6. Figure 5 and 7, It will be better to provide a schematic illustration similar to Figure 2A.

Thank you for the comment. Schematic illustrations have been added in Fig 5A and Fig 7A.

7. Figure 7D, what about other M1/M2 markers (CD86, MHCII, CD206)?

Thank you. Compared with IL-33 monotherapy group, the proportion of CD206⁺ ($P = 0.0033$, Appendix Fig S11A) macrophages decreased, and the proportion of CD86⁺ ($P = 0.0048$, Appendix Fig S11B) and MHC-II⁺ ($P = 0.0273$, Appendix Fig S11C) macrophages increased in the combined treatment group. We have added and highlighted the data in Results section (p19 of revised manuscript). Appendix Fig S11 is as follows.

8. Please provide scientific support for the dose/duration of IL-33, anti-CSF1R and SB203580.

Thank you for the comment.

(1) The dose and duration of IL-33, anti-CSF1R and SB203580 *in vivo* are determined according to previous studies. We have added and highlighted the explanations and references in the Materials and Methods section (p37 of revised manuscript). The references are as follows:

Dalmas E, Lehmann FM, Dror E, et al. Interleukin-33-Activated Islet-Resident Innate Lymphoid Cells Promote Insulin Secretion through Myeloid Cell Retinoic Acid Production. *Immunity*. 2017 Nov 21;47(5):928-942.e7. PMID: 29166590.

Barkal AA, Brewer RE, Markovic M, et al. CD24 signalling through macrophage Siglec-10 is a target for cancer immunotherapy. *Nature*. 2019 Aug;572(7769):392-396. PMID: 31367043.

Leelahavanichkul K, Amornphimoltham P, Molinolo AA, et al. A role for p38 MAPK in head and neck cancer cell growth and tumor-induced angiogenesis and lymphangiogenesis. *Mol Oncol*. 2014 Feb;8(1):105-18. PMID: 24216180.

(2) The dose of IL-33 and SB203580 *in vitro* are also based on previous studies. We have added the explanations and references in the Materials and Methods section (p35 of revised manuscript). The references are as follows:

Yang Y, Andersson P, Hosaka K, et al. The PDGF-BB-SOX7 axis-modulated IL-33 in pericytes and stromal cells promotes metastasis through tumour-associated macrophages. *Nat Commun*. 2016 May 6;7:11385. PMID: 27150562.

Zhou MM, Zhang WY, Li RJ, et al. Anti-inflammatory activity of Khayandirobilide A from *Khaya senegalensis* via NF- κ B, AP-1 and p38 MAPK/Nrf2/HO-1 signaling pathways in lipopolysaccharide-stimulated RAW 264.7 and BV-2 cells. *Phytomedicine*. 2018 Mar 15;42:152-163. PMID: 29655681.

Thank you again for your constructive suggestions.

5th Sep 2023

Dear Jia,

Congratulations on a great revision! Overall, the referees have been positive. However, one referee has a few remaining concerns that we ask you to address in a new revision. When you submit your revised version, please also take care of the following editorial items and add this also to a new point-by-point response:

1. We found that not all of the data listed in the Data availability section is accessible, please ensure readers have access to this data.
2. Please review our policy on conflict of interests on the EMM author guide website and update the title of this section to: Disclosure and competing interests statement.
3. For references, the title of all publications should be italicized and attention should be paid to the use of upper case letters.
4. The heading "additional information" and the section "consent for publication" should be removed.

Thank you for the opportunity to consider your work for publication. I look forward to your revision.

Kind regards,
Kelly

Kelly M Anderson, PhD
Scientific Editor
EMBO Molecular Medicine

*** Instructions to submit your revised manuscript ***

To submit your manuscript, please follow this link:

<https://embomolmed.msubmit.net/cgi-bin/main.plex>

- 1) a .docx formatted version of the manuscript text (including Figure legends and tables)
- 2) Separate figure files*
- 3) supplemental information as Expanded View and/or Appendix. Please carefully check the authors guidelines for formatting Expanded view and Appendix figures and tables at <https://www.embopress.org/page/journal/17574684/authorguide#expandedview>
- 4) a letter INCLUDING the reviewer's reports and your detailed responses to their comments (as Word file).
- 5) The paper explained: EMBO Molecular Medicine articles are accompanied by a summary of the articles to emphasize the major findings in the paper and their medical implications for the non-specialist reader. Please provide a draft summary of your article highlighting
 - the medical issue you are addressing,
 - the results obtained and

- their clinical impact.

This may be edited to ensure that readers understand the significance and context of the research.

Please refer to any of our published articles for an example.

6) For more information: There is space at the end of each article to list relevant web links for further consultation by our readers. Could you identify some relevant ones and provide such information as well? Some examples are patient associations, relevant databases, OMIM/proteins/genes links, author's websites, etc...

7) Author contributions: the contribution of every author must be detailed in a separate section.

8) EMBO Molecular Medicine now requires a complete author checklist

(<https://www.embopress.org/page/journal/17574684/authorguide>) to be submitted with all revised manuscripts. Please use the checklist as guideline for the sort of information we need WITHIN the manuscript. The checklist should only be filled with page numbers where the information can be found. This is particularly important for animal reporting, antibody dilutions (missing) and exact values and n that should be indicated instead of a range.

9) Every published paper now includes a 'Synopsis' to further enhance discoverability. Synopses are displayed on the journal webpage and are freely accessible to all readers. They include a short stand first (maximum of 300 characters, including space) as well as 2-5 one sentence bullet points that summarise the paper. Please write the bullet points to summarise the key NEW findings. They should be designed to be complementary to the abstract - i.e. not repeat the same text. We encourage inclusion of key acronyms and quantitative information (maximum of 30 words / bullet point). Please use the passive voice. Please attach these in a separate file or send them by email, we will incorporate them accordingly.

You are also welcome to suggest a striking image or visual abstract to illustrate your article. If you do please provide a jpeg file 550 px-wide x 400-px high.

10) A Conflict of Interest statement should be provided in the main text

11) Please note that we now mandate that all corresponding authors list an ORCID digital identifier. This takes <90 seconds to complete. We encourage all authors to supply an ORCID identifier, which will be linked to their name for unambiguous name identification.

Currently, our records indicate that the ORCID for your account is 0000-0003-3024-8878.

Link Not Available

12) The system will prompt you to fill in your funding and payment information. This will allow Wiley to send you a quote for the article processing charge (APC) in case of acceptance. This quote takes into account any reduction or fee waivers that you may be eligible for. Authors do not need to pay any fees before their manuscript is accepted and transferred to our publisher.

Photos 400-800 DPI

*Additional important information regarding figures and illustrations can be found at

<https://bit.ly/EMBOPressFigurePreparationGuideline>. See also figure legend preparation guidelines:

<https://www.embopress.org/page/journal/17574684/authorguide#figureformat>

The system will prompt you to fill in your funding and payment information. This will allow Wiley to send you a quote for the article processing charge (APC) in case of acceptance. This quote takes into account any reduction or fee waivers that you may be eligible for. Authors do not need to pay any fees before their manuscript is accepted and transferred to our publisher.

***** Reviewer's comments *****

Referee #1 (Comments on Novelty/Model System for Author):

As I have stated in my comments, at least two points, the authors can't or not willing to do so, but only referred to previous publications.

Referee #1 (Remarks for Author):

I have asked the questions:

1. Any expression of MCSF-1R in GC? 2. What is the half-life of IL-33 and anti MCSF-1R? The authors didn't perform the tests, but only referring the publications. I think it isn't sufficient.

3. I was asking why M2 macrophage polarisation inhibiting the development of gastric cancer cell, but the answer wasn't satisfactory.

As I have commented previously, the manuscript is too long. I don't think the authors understood that this is a manuscript not a thesis. Much of the contents in the Materials and Methods should be presented concisely.

Referee #3 (Comments on Novelty/Model System for Author):

I have no further comments.

Referee #3 (Remarks for Author):

I have no further comments.

Referee #1

1. Any expression of MCSF-1R in GC?

Thank you. IHC analysis of MCSF-1R was performed on the specimens of primary GC. We found that MCSF-1R was expressed in GC cells. Scale bar, 50 μ m.

2. What is the half-life of IL-33 and anti MCSF-1R?

Thank you for your comment. We performed the test to evaluate the *in vivo* half-life of IL-33 in mice. 615-line mice were injected intraperitoneally with IL-33 (20 µg/kg). Serum was collected at various times after injection and IL-33 levels were measured by ELISA. IL-33 reached peak plasma level at 30 min after protein administration. The half-life of IL-33 was 1 h.

About the half-life of anti-MCSF-1R, we have studied a lot of relevant literatures and designed our experiments with reference to previous similar studies. The dose and duration of anti-MCSF-1R were used as previous described by Barkal et al. in *Nature* (The reference is as follows). In our study, the infiltrating TAMs (CD11b⁺ and F4/80⁺) were assessed to ensure macrophage depletion after the administration of anti-MCSF-1R (Figure is as follows). Therefore, further testing about the half-life of anti-MCSF-1R does not affect the results of our study. If the editor deems it necessary, we may need some time to perform the test. Thank you for your advice.

Barkal AA, Brewer RE, Markovic M, et al. CD24 signalling through macrophage Siglec-10 is a target for cancer immunotherapy. *Nature*. 2019 Aug;572(7769):392-396. PMID: 31367043.

3. I was asking why M2 macrophage polarization inhibiting the development of gastric cancer cell, but the answer wasn't satisfactory.

Thank you. M2 macrophages polarization can not inhibit the development of gastric cancer. On the contrast, M2 macrophages promote the progression of gastric cancer. Our study illustrates the dual role of IL-33. The increase of IL-33 levels locally resulted in the recruitment of CD8⁺T cells, NK cells, DCs and increased the expression of IFN- γ in CD8⁺ T cells and NK cells, indicating the activation of immunologic effector cells, which plays an anti-tumor role. However, IL-33 also promotes the recruitment of TAMs and M2 polarization of macrophages to a certain extent, which has an immunosuppressive effect. Thus, the combination of IL-33 and anti-CSF1R or p38 inhibitor could avoid the negative immune regulation of IL-33-induced M2 polarization, which has synergistic anti-tumor effect.

4. As I have commented previously, the manuscript is too long. I don't think the authors understood that this is a manuscript not a thesis. Much of the contents in the Materials and Methods should be presented concisely.

Thank you. We have modified the Materials and Methods and removed redundant parts. In order to present concisely, commercially available antibodies and reagents used in this study were listed in Appendix Table S2 and S3. Thank you again for your kind advice.

Editorial items

1. We found that not all of the data listed in the Data availability section is accessible, please ensure readers have access to this data.

Thank you. To respect patient confidentiality, the clinicopathological data of patients can be obtained from the corresponding author. We will assist readers in obtaining data access agreement with Nanjing Drum Tower Hospital to ensure readers have access to this data.

2. Please review our policy on conflict of interests on the EMM author guide website and update the title of this section to: Disclosure and competing interests statement.

Thank you. We have updated the title of conflict of interests section to: Disclosure and competing interests statement.

3. For references, the title of all publications should be italicized and attention should be paid to the use of upper case letters.

Thank you. The titles of all publications have been italicized and used upper case letters.

4. The heading "additional information" and the section "consent for publication" should be removed.

Thank you. we have removed the heading "additional information" and the section "consent for publication".

20th Nov 2023

Dear Jia,

Congratulations on an excellent manuscript, I am pleased to inform you that your manuscript has been accepted for publication in the EMBO Journal. Thank you for your comprehensive response to the referee concerns and for providing detailed source data. It has been a pleasure to work with you to get this to the acceptance stage.

I will begin the final checks on your manuscript before submitting to the publisher next week. Once at the publisher, it will take about three weeks for your manuscript to be published online. As a reminder, the entire review process, including referee concerns and your point-by-point response, will be available to readers.

I will be in touch throughout the final editorial process until publication. In the meantime, I hope you find time to celebrate!

Warm wishes,
Kelly

Kelly M Anderson, PhD
Scientific Editor
EMBO Molecular Medicine
